# Visual Representations inside the Language Model

**Benlin Liu[1], Amita Kamath[1,2], Madeleine Grunde-McLaughlin[1],**
**Winson Han[3], Ranjay Krishna[1,3]**
[1]University of Washington [2]University of California Los Angeles [3]Allen Institute for AI
liubl@cs.washington.edu

## Abstract

Despite interpretability work analyzing VIT encoders and transformer activations, we don't yet understand why Multimodal Language Models (MLMs) struggle on perception-heavy tasks. We offer an under-studied perspective by examining how popular MLMs (LLaVA-OneVision, Qwen2.5-VL, and Llama-3-LLaVA-NeXT) process their visual key-value tokens. We first study the flow of visual information through the language model, finding that image value tokens encode sufficient information to perform several perception-heavy tasks zero-shot: segmentation, semantic correspondence, temporal correspondence, and referring expression detection. We find that while the language model does augment the visual information received from the projection of input visual encodings—which we reveal correlates with overall MLM perception capability—it contains less visual information on several tasks than the equivalent visual encoder (SigLIP) that has *not* undergone MLM finetuning. Further, we find that the visual information corresponding to input-agnostic image key tokens in later layers of language models contains artifacts which *reduce* perception capability of the overall MLM. Next, we discuss controlling visual information in the language model, showing that adding a text prefix to the image input improves perception capabilities of visual representations. Finally, we reveal that if language models *were* able to better control their visual information, their perception would significantly improve; e.g., in 33.3% of Art Style questions in the BLINK benchmark, perception information present in the language model is not surfaced to the output! Our findings reveal insights into the role of key-value tokens in multimodal systems, paving the way for deeper mechanistic interpretability of MLMs and suggesting new directions for training their visual encoder and language model components.

## 1 Introduction

Multimodal language models (MLMs) still struggle on perception tasks (Fu et al., 2024; Tong et al., 2024b), particularly those that require reasoning over relative depth, object localization, identifying object segments, and spatial understanding (Kamath et al., 2023; Hu et al., 2024). Contemporary studies indicate that such perceptual capabilities can be improved with specialized task-designs or datasets (Ray et al., 2024; Bigverdi et al., 2025). Despite empirical improvements on benchmarks with specialized designs, we lack the scientific tools to dissect what causes these improvements. We conjecture that the answer to this question requires understanding how MLMs represent and reason over visual inputs.

Efforts to interpret vision-language models have traditionally focused on two main directions. The first direction analyzes ViT-based encoders such as CLIP (Radford et al., 2021) and DINO (Caron et al., 2021), aiming to understand how visual features are extracted and mapped into semantic space (Dosovitskiy et al., 2020). For instance, one method identifies the role of each attention head within CLIP, then reduces spurious correlations by removing specific attention heads (Gandelsman et al., 2024). The second direction examines the outputs of the transformer blocks that process visual tokens in MLMs (Liu et al., 2024; Wang et al., 2024), e.g., finding that visual representations gradually align with textual

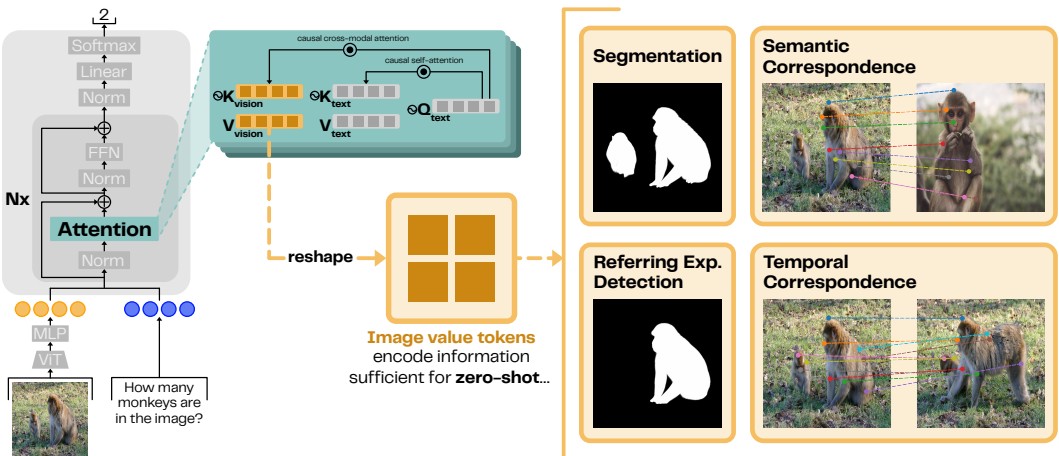

Figure 1: We study visual representations in the key-value cache within Mutimodal Language Models (MLMs), as they are uninfluenced by text (due to the causal nature of cross-modal attention) and directly contribute to the MLM output. Despite MLMs struggling on perception (Fu et al., 2024; Tong et al., 2024b), we find that their intermediate image value tokens encode sufficient information for various zero-shot perception tasks—calling for research to understand and surface the visual information already present within MLMs.

concepts across layers (Neo et al., 2024). While insightful, these two directions overlook a critical feature that distinguishes MLMs from traditional ViTs: how visual information is represented and used inside the language model.

We argue that the key-value (KV) tokens corresponding to the visual inputs are a critical—and understudied—lens for understanding how language models process visual information. In MLMs, an image is typically encoded by an encoder (with an optional lightweight projection) and then fed into a language model as visual tokens. Cross-modal attention draws on keys and values to determine what information to retrieve about an image. Analyzing these KV tokens can clarify how MLMs integrate visual details alongside linguistic context. Although interpretability approaches for text-only language models have studied the KV cache (Ge et al., 2024), they focus on improving efficiency and do not translate over when studying visual tokens inside a language model. Moreover, the cache may reveal factors that either enhance or degrade visual fidelity in MLMs, giving us handles to improve perception-oriented tasks.

We investigate the internal KV representations of three popular and distinct models: LLaVA-OneVision 7B (Li et al., 2025a), Qwen2.5-VL 7B (Bai et al., 2025), and Llama-3-LLaVA-NeXT 8B (Li et al., 2025b). We first find that the models' image value tokens encode sufficient information to perform zero-shot perception-heavy tasks (foreground segmentation, semantic segmentation, co-segmentation, referring expression segmentation, semantic correspondence, and temporal correspondence), and that their performance on these tasks correlates with perception performance of the overall MLM. We then use these tasks to measure perception capability of the visual representations in different layers of the MLM, shedding light on the flow of visual information through the language model. We find that, similar to semantic information in text-only language models, the segmentation capability of the visual representations gradually builds over the first two-third layers, then drops off steeply in the later layers, likely as the model begins to focus on generating the text output.

However, while the language model layers do initially augment the input visual information received from the projection of the input visual encodings, we show that on several tasks, they have less perception capability than encodings from the equivalent visual encoder (SigLIP) that has *not* undergone MLM finetuning, which agrees with findings from concurrent work (Fu et al., 2025). Further, we find that visual information corresponding to input-agnostic image key tokens in later layers of language models contains artifacts which actually *reduce* perception capability of the overall MLM.

Having built an understanding of the visual representations in the language model and the impact of the visual information they contain on perception capabilities of the MLM, we next discuss *controlling* the visual information within the language model. We first introduce a promising method to improve this visual information: adding a textual prefix to the image. The promptable nature of visual representations in the language model enables them to dynamically adapt to the prefix via causal attention, which we show improves their perceptual capabilities on three tasks: referring expression segmentation, semantic correspondence and domain adaptation for zero-shot semantic segmentation. Finally, we reveal that if language models *were* able to better control the visual information they contain, their perception capabilities would significantly improve; e.g., in 33.3% of Art Style questions in the BLINK benchmark (Fu et al., 2024), the correct perception information is present in the language model, but is not surfaced to the final output.

Our work lays the foundation for further studies in mechanistic interpretability of visual representations in MLMs, as well as how to leverage the same to improve model performance on perception-heavy tasks—particularly in the cases we reveal, where the visual information is already present within the model. Our findings also suggest new directions for training both the visual encoder and language model components of MLMs.

## 2 Preliminaries

MLMs typically consist of three main components: an image encoder, a lightweight adapter, and a language decoder. The image encoder, often a ViT backbone, processes raw image inputs into a set of embeddings. The connector, usually a linear layer, then projects these embeddings into a space compatible with the language model. Finally, the language decoder, typically a Transformer-based autoregressive model, integrates both textual and visual information to generate responses. In this section, we discuss how images are processed by the visual encoder, and contrast it with how they are processed by the language model.

### 2.1 Image processing by the visual encoder

Each image is divided into $N$ patches, flattened, and projected into $D$-dimensional embeddings. After adding positional encodings, these embeddings form a sequence: $X = [X_1, X_2, ..., X_N]$, where $X_i$ is the representation of the $i^{\text{th}}$ patch. These patch embeddings are then passed into a Multi-Head Self-Attention mechanism (Vaswani et al., 2017).

Each patch representation $X_i$ is linearly transformed into three separate vectors: Query ($Q$), Key ($K$), and Value ($V$). This is done using learnable weight matrices: $Q = XW_Q, K = XW_K, V = XW_V$, where $W_Q, W_K, W_V$ are learned weight matrices of shape $D \times d_k$ and $Q, K, V$ are of shape $N \times d_k$, where $d_k$ is the dimension of each attention head.

Self-attention is then computed using the scaled dot-product attention mechanism:

$$\text{Attention}(Q, K, V) = \text{softmax}\left(\frac{Q(K)^T}{\sqrt{d_k}}\right) V.$$

ViT uses a multi-head attention mechanism in which $h$ self-attention operations, called "heads", are run in parallel and concatenated before projection to the output space. Following the attention block, the token representations are passed through a feedforward network (FFN), usually consisting of a two-layer MLP with a nonlinearity between layers. Layer normalization and residuals are applied before and after attention and FFN respectively.

Every visual token attends to every other token in a dense global self-attention mechanism. As ViTs compute $Q, K, V$ on the fly for every token in a fully tensorized operation, they do not store representations separately for later querying.

### 2.2 Image processing within the language model

The key differences between image processing in MLMs versus ViT are: (1) an explicit key-value storage mechanism; (2) causal attention; and (3) cross-attention between modalities. We detail each below, along with how they relate to text generation by the language model.

**Explicit key-value storage.** In standard transformers, a key-value (KV) cache is used to efficiently store and retrieve token embeddings during self-attention: rather than recomputing attention scores for every token at every step, transformers store keys and values for previously seen tokens. In MLMs, this is operationalized by storing pre-computed visual information as static key-value tokens in the cache, allowing text tokens to selectively retrieve them via causal cross-modal attention.

**Causal attention.** In autoregressive decoding, a causal mask $M$ is applied, preventing tokens from attending to future positions:

$$\text{Attention}(Q, K, V) = \text{softmax}\left(\frac{Q(K)^T}{\sqrt{d_k}} + M\right) V$$

In MLMs, the image is generally positioned before the text in the input. Hence, causal attention ensures that text tokens only attend to previous text and visual tokens, and that visual tokens do not attend to text tokens (i.e., there is one-way querying), allowing them to be pre-computed and stored statically in the KV cache (c.f. Appendix for details).

**Cross-modal attention.** Due to the positioning of images before text in the input, the LLM can query visual features when generating text via causal attention, i.e., they can "look up" relevant visual information: $\text{Attention}(Q_{\text{text}}, K_{\text{vision}}, V_{\text{vision}})$. Vision tokens do not influence text generation directly, i.e., they do not modify the text keys or values (similar to how a text token does not modify successive text tokens in the input).

**Group query attention.** In our experiments, we use models that adopt the *Group Query Attention (GQA)* (Ainslie et al., 2023) mechanism. Unlike standard multi-head attention, where each attention head has its own independent query, key, and value projections, GQA shares keys and values across heads while maintaining distinct queries. Specifically, all attention heads in a group share the same key and value matrices but compute attention independently using their own query projections (c.f. Appendix for details):

$$\text{head}_i = \text{Attention}(Q_i, K_{\text{shared}}, V_{\text{shared}})$$

**Text generation by the language model.** The weighted sum of values omitted by the attention mechanism is passed through the remainder of the transformer block to the next layer, successively until the final layer, when it is normalized, passed through a linear layer and softmax, and generates a prediction probability distribution over the vocabulary, predicting each token of the output auto-regressively.

To understand visual representations within the language model, we study the image-based keys and values in the attention mechanism (c.f. Figure 1). First, due to causal attention and the typical position of image before text, these key-value pairs are unaffected by the text input and thus form a **purely visual representation**. Second, once computed, they are stored and fixed, allowing all text tokens to attend to image keys and retrieve values that **directly contribute to the output**. As such, they serve as a *proxy for the input image*, remaining accessible throughout any downstream language task. Understanding what is encoded in these representations is therefore essential for improving perception in MLLMs.

## 3 Visual Information within the Language Model

In this section, we study how visual information flows through the language model, as well as its perceptual capability as measured by six perception-heavy probing tasks. We discuss how this capability compares to popular visual encoders, and that it correlates with overall MLM performance on perception-heavy tasks. We then reveal that certain visual information in the language model actually degrades MLM perception.

### 3.1 Flow of visual information through the language model

Despite the use of strong vision encoders (Zhai et al., 2023), MLMs continue to perform poorly on vision-centric benchmarks (Fu et al., 2024; Tong et al., 2024b). This suggests that the language model may not correctly process visual information passed into it by the vision

| | Segmentation | | | | Correspondence | | | |
| | Pascal-5i (mIOU) | MSRC ($\mathcal{J}_m$) | ImageNet-S (mIOU) | RefCOCO (mIOU) | SPair71K (PCK) | DAVIS ($\mathcal{J}\&\mathcal{F}_m$) | ($\mathcal{J}$) | ($\mathcal{F}_m$) |
|---|---|---|---|---|---|---|---|---|
| Visual Encoder (post-proj) | 24.2 | 58.4 | 51.1 | 59.6 | 36.9 | 57.4 | 56.8 | 58.0 |
| LM Image Value (max) | 26.2 | 61.6 | 54.6 | **64.8** | **46.1** | **65.8** | **62.6** | **67.2** |
| SigLIP *without* MLM FT | **27.3** | **62.0** | **59.1** | 61.7 | 41.6 | 59.8 | 58.9 | 60.7 |

Table 1: Performance of different visual representations on segmentation and correspondence probing tasks. The highest-performing image value in the LM outperforms the input visual representation from the visual encoder (post-projection), but falls short of SigLIP without MLM finetuning on the first three segmentation tasks.

encoder. We study the flow of visual information through the language model by probing the image *value* tokens[1] in the KV cache of each layer of the language model. We discuss LLaVA-OneVision 7B in this section, and equivalent experiments for Qwen2.5-VL 7B and Llama-3-LLaVA-NeXT 8B in the Appendix. Our observations are consistent across models.

**Why values?** As discussed in Section 2.2, image value tokens serve as a proxy for the input image at each layer within the language model. Further, individual image values contribute to MLM language generation by aggregating spatially distributed information, per the position encoding added to the corresponding image key. Therefore, image values should encode well-localized semantics of the image, while maintaining cross-modal alignment.

**Six probing tasks.** To assess whether image values contain aforementioned visual information, we probe them on 6 tasks requiring the same. We discuss below each task alongside its experimental setup, where all tasks are performed **zero-shot** unless otherwise specified.

*Few-shot foreground segmentation.* We evaluate on Pascal-5i (split 0) (Shaban et al., 2017). We train a linear classifier on top of each image value to perform 5-shot binary segmentation.

*Co-segmentation.* We evaluate on MSRC (Shotton et al., 2006), which contains sets of images across which the model must segment a similar object. For each image value token, we follow Amir et al. (2022) and perform spatial clustering to segment objects across images.

*Semantic segmentation.* We evaluate on ImageNet-S (Gao et al., 2022). We project the text of the class name into the representational space of the image value tokens using the MLM (independent of the image), then choose the final text token as the class representation. We select image values with highest dot product with this text token as the segmented region.

*Referring expression segmentation.* We evaluate on RefCOCO (Kazemzadeh et al., 2014), which requires grounding language to an object in the image. We follow the same method as semantic segmentation, but with text of the referring expression rather than the class name.

*Semantic correspondence.* We evaluate on SPair71K (Min et al., 2019), in which models must establish semantic correspondences between keypoints annotated on pairs of images. We find nearest neighbors between images for each image value corresponding to a keypoint.

*Temporal correspondence.* We evaluate on DAVIS (Pont-Tuset et al., 2017), in which models propagate the mask of an object across video frames. We follow the same method as for semantic correspondence.

**Image value tokens are capable of perception-heavy tasks.** We evaluate each image value token in the language model (i.e., across all heads and layers), and report the highest performance among them in Table 1 as "LM Image Value (max)". These visual representations show the capability to perform a wide variety of perception tasks, from different types of segmentation to different types of correspondence across images.

**Visual information is refined in the language model.** We compare the results of the highest-performing image value token in the language model to the performance of the input visual representation received by the language model, i.e., the representation produced by

---

[1]We use "image {key, value} token" interchangeably with "image {key, value}".

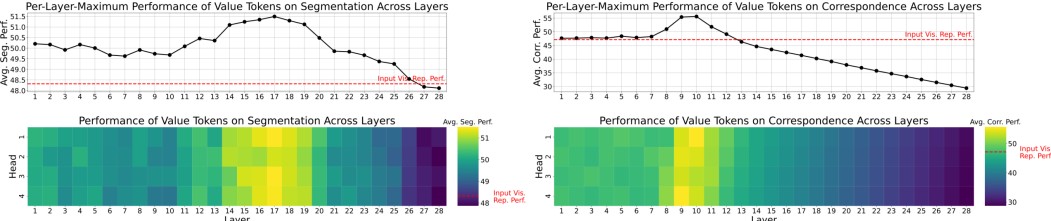

Figure 2: Performance of each image value (bottom), and maximum per-layer image value performance (top). On segmentation tasks (left), visual information builds gradually in the first two-thirds of the language model (LM), then drops steeply. On correspondence tasks (right), visual information builds sharply after the first third of the LM, then drops steadily (note the scale of the Y-axis). The LM builds upon the input visual representation (in red).

| Visual | SPair-71K | DAVIS | | |
| Representation | PCK | $\mathcal{J}\&\mathcal{F}_m$ | $\mathcal{J}_m$ | $\mathcal{F}_m$ |
|---|---|---|---|---|
| CLIP | 41.4 | 62.5 | 60.6 | 64.4 |
| DINO | 36.7 | **71.4** | **67.9** | **74.9** |
| LM Image Value (max) | **46.1** | 65.8 | 62.6 | 67.2 |

Table 2: Performance of visual representations on correspondence probing tasks. The highest-performing image value in the LM compares favorably to strong vision encoders.

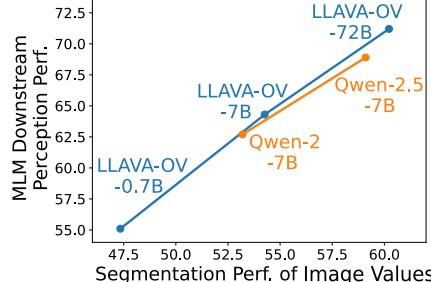

Figure 3: Segmentation performance of image values correlates with MLM perception on downstream tasks.

the projection layer after the visual encoder. The highest-performing image value token outperforms the input visual representation across all six probing tasks. This shows that the language model refines the visual information it receives, building upon the information contained by the visual representations.

**Visual information builds, then drops, across language model layers.** We now study how visual information evolves across the layers of the language model. Figure 2 (bottom) depicts the average segmentation and correspondence performance of the image value tokens corresponding to each head in each layer of the language model. Figure 2 (top) tracks the maximum performance per layer. For the segmentation task, the early layers (1–15) gradually refine visual information, which peaks at layers 14–15, after which it drops off steeply, likely as later layers of the model begin to focus on generating the text output. This is similar to the processing of semantic information by language models (Tenney et al., 2019). For the correspondence task, the information is initially retained, then refined in layers 7–10, after which it drops off steadily. This is likely due to the fact that the correspondence task abstracts away pixel appearance, which is needed for segmentation and for most multimodal tasks. c.f. Appendix for per-task observations.

**Visual information of the MLM visual encoder is less than that of the equivalent visual encoder (SigLIP) *without* MLM finetuning.** LLaVA-OneVision 7B uses SigLIP v1-384 as its visual encoder, finetuning it during multimodal training. We compare the performance of the original SigLIP v1-384 model with the finetuned visual encoder in LLaVA-OneVision (post-projection), to determine how the visual information captured by the visual encoder changes during the MLM finetuning process. As shown in Table 1, we find that the original SigLIP model actually outperforms the finetuned visual encoder across all 6 probing tasks. Further, while the language model refines the visual information it receives from the fine-tuned visual encoder, it underperforms the original SigLIP model on 3 of 6 probing tasks (foreground segmentation, co-segmentation, and semantic segmentation)—hinting that the language model is not always able to compensate for the lost information. This agrees with concurrent observations that MLMs can under-perform visual encoders (Fu et al., 2025).

Images      Feature maps of keys

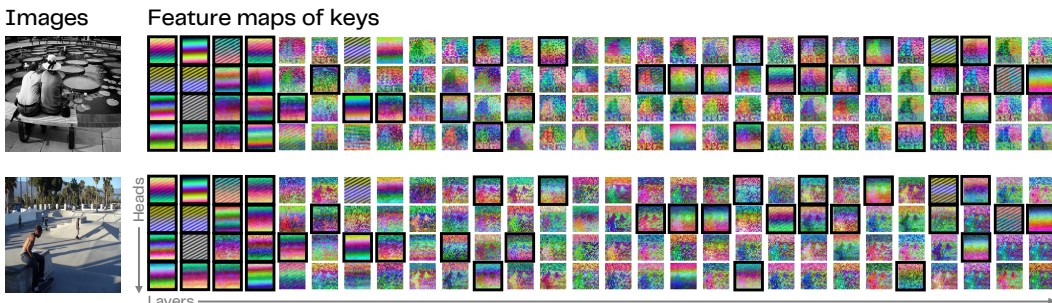

Figure 4: PCA visualization of all image keys in LLaVA-OneVision 7B for two COCO images. The input-agnostic image keys (highlighted) alone are nearly constant across images.

**Visual information in the language model compares favorably to that of other visual encoders.** In Table 2, we compare the results of the highest-performing image value in the language model in LLaVA-OneVision 7B to the performance of CLIP (Radford et al., 2021) and DINO (Caron et al., 2021) visual representations on our correspondence probing tasks. We find that the highest-performing image value outperforms both CLIP and DINO on semantic correspondence, and outperforms CLIP on temporal correspondence (but not DINO). This can be partially attributed to the benefits of cross-modality of the image value, and partially attributed to the training procedure of LLaVA-OneVision, which includes multi-image and video data—similar to the pairs of images in correspondence tasks.

**Visual information in the language model correlates with downstream perception performance of the MLM.** We assess MLMs' end-to-end perception reasoning by evaluating them on perception-heavy tasks in CV-Bench (Tong et al., 2024b) and BLINK (Fu et al., 2024), and compare their performance to that of their highest-performing image value tokens averaged across the segmentation probing tasks. As shown in Figure 3, we observe that MLMs of different sizes and versions exhibit the same trend: the stronger the perception capability of their image value tokens, the better the MLMs themselves perform on downstream perception-heavy tasks. This correlation highlights the importance of visual information within the language model. Further details about this experiment are in the Appendix.

### 3.2 Certain visual information in the language model *degrades* MLM perception

Having studied the perception capability of visual representations in the language model, we now ask: why do MLMs still perform poorly on perception-heavy tasks? We show, via an intervention study, that certain visual information in the language model actually *degrades* perception capability of the MLM. We focus on LLaVA-OneVision 7B in this section, and discuss equivalent experiments for Qwen2.5-VL 7B in the Appendix. Our observations are consistent across models.

Our study is inspired by Darcet et al. (2024), who reveal that in ViTs, tokens in middle layers corresponding to background patches tend to encode input-agnostic artifacts rather than useful information about the image. We investigate whether a similar phenomenon occurs in MLMs by identifying input-agnostic visual representations in the language model, and determining whether they, too, encode artifacts instead of helpful visual information.

**Input-agnostic image key tokens dominate in early layers of the language model, and re-emerge in late laters.** We sample 1000 images from MSCOCO (Lin et al., 2014) and obtain intermediate visual representations for each sample from LLaVA-OneVision 7B. We compute the variance of image key tokens in each head and layer across the samples. We find that the variance forms a bimodal distribution, which we split with a threshold to define input-agnostic keys (details in Appendix). As shown in Figure 4, we find that when setting this threshold, input-agnostic keys dominate in earlier layers of the language model, but, critically, re-emerge in later layers. We test whether these input-agnostic keys in later layers too, contain input-agnostic artifacts as in ViTs with an intervention study.

**Blocking visual information corresponding to input-agnostic image key tokens in later layers improves MLM perception.** We conduct an interventional study, blocking text token queries to input-agnostic image keys in the last 10 layers via attention knockout (Geva et al., 2023). Notably, we do *not* block text keys that belong to the same attention head as these input-agnostic image keys, ensuring a focus on understanding the impact of visual representations on model output. We evaluate on two benchmarks: POPE (Li

| Model | POPE | MME* |
|---|---|---|
| LLaVA-OneVision 7B | 78.1 | 173.3 |
| + intervention | **81.2** | **179.0** |

Table 3: Our intervention reveals image keys that contain artifacts, as blocking them improves MLM performance on POPE and MME.

et al., 2023), and 4 subsets of MME (Fu et al., 2023) which together address hallucination and general perception capabilities. As shown in Table 3, we find that blocking image-agnostic image keys (and thus, image values) in later layers improves model performance on both benchmarks. In the Appendix, we show that blocking these keys in early layers significantly reduces model performance, and discuss experimental controls. By showing that removing the influence of these visual representations on the language output improves the MLM performance, we prove that image values corresponding to input-agnostic image keys in later layers of the language model likely encode artifacts rather than helpful visual information.

We posit that input-agnostic image keys in earlier layers play an essential role in aggregating contextual information consistent across different inputs, whereas middle-to-later layers of the model gradually shift focus toward understanding semantics (Neo et al., 2024), similar to CNNs (Krizhevsky et al., 2012) and pure-text language models (Tenney et al., 2019). Hence, the input-agnostic image keys in later layers of the model are likely to encode artifacts, similar to ViTs (Darcet et al., 2024). Our findings shed light on visual information in the language model that degrades MLM perception, calling for further research in identifying why this phenomenon occurs, and how to mitigate the same.

## 4 Controlling Visual Information in the Language Model

Thus far, we have discussed how visual information flows through the language model, and how it impacts MLM perception performance positively and negatively. In this section, we discuss controlling this visual information: leveraging the language model mechanism of causal attention to improve visual information via textual prefixing, and then revealing the significant scope for improvement in MLM perception if language models *could* better control their visual information.

### 4.1 Improving visual information by leveraging causal attention in language models

Visual representations in the language model have an interesting capability thanks to their cross-modal nature: they can be made promptable by prefixing textual prompts to the visual input. Causal cross-modal attention then allows the model to dynamically adapt its visual representations on the fly to better align with specific tasks. We study this in the context of three tasks: referring expression segmentation, semantic correspondence, and domain adaptation for semantic segmentation. We present results in Table 4.

**Prefixing improves referring expression segmentation.** We evaluate image values from LLaVA-OneVision 7B on RefCOCO. Without text prefixing, the experiment is as described in Section 3.1. When a textual prefix is added to the image input (here, the image caption from COCO (Chen et al., 2015)), the image values achieve modestly improved segmentation results. This shows that the visual representations were able to incorporate the text caption—which contains information about the image, and may contain information about the object to segment—to improve their textual grounding and segmentation capabilities.

**Prefixing improves semantic correspondence.** We evaluate image value tokens from LLaVA-OneVision 7B on SPair71K. Without text prefixing, the experiment is as described in Section 3.1. The textual prefix we add for this task is the class name of the object in the image. As with referring expression segmentation, the semantic correspondence performance improves slightly with prefixing, showing that the visual representations incorporate textual information from the prefix to improve textual grounding and segmentation capabilities.

| | RefCOCO | SPair71K | BDD-10K day+street | BDD-10K night+highway |
|---|---|---|---|---|
| No prefixing | 64.8 | 46.1 | 69.7 | 63.4 |
| Prefixing | **65.3** | **46.5** | **70.4** | **68.2** |
| Prefixing-random | 64.4 | 45.8 | 68.9 | 63.2 |
| Prefixing-incorrect | - | 44.9 | 68.5 | 62.1 |

Table 4: Performance of image values on segmentation and correspondence tasks improves with prefixing of a relevant text prompt (top), and not a random or incorrect one (bottom).

**Prefixing improves domain adaptation for zero-shot semantic segmentation.** We evaluate image values from LLaVA-OneVision 7B on BDD-10K (Yu et al., 2020) on two domains: city street during daytime, and highway during nighttime. Without prefixing, the experiment is similar to semantic segmentation described in Section 3.1. By prefixing the image with text specifying the domain ("The image is taken at {daytime, nighttime} and it is from {city street view, highway}"), performance improves on both domains, but more significantly on the more challenging one. This shows that the visual representations successfully incorporate the text information identifying the domains, then adapt on the fly to each.

**Controls.** We run two controls for this experiment, to ensure the *content* of the text prefix is what improves the visual representation quality, not the addition of a text prefix itself. For all tasks, we prefix a random string of text ("A rustic wooden table filled with freshly baked croissants, dripping honey, and a steaming pot of Earl Grey tea beside a bowl of ripe figs.") to the image and calculate performance ("Prefixing-random"). Across all tasks, this does not improve performance compared to no prefixing, and rather lowers it slightly. For the semantic correspondence task, we add a second control wherein we prefix the name of the *incorrect* class ("Prefixing-incorrect"). We do the same for domain adaptation, in which we prefix the name of the *incorrect* domain ("The image is taken at daytime and it is from city street view" for an image of night+highway, and vice versa). These numbers are even lower than the random prefix for both tasks. These results collectively show that it is indeed the prefix content that improves visual representation quality.

These results suggest that image value tokens are not only effective for visual understanding, but are also inherently adaptable to linguistic conditioning. Further, they set forth a promising method to improve visual information by leveraging mechanisms already present in the language model. Our findings call for further research in improving control of visual information to improve perception capabilities within the language model.

## 4.2 Better control of visual information in the language model could improve MLM perception *significantly*

We have shown in Section 3.1 that visual representations in the language model are capable of perception-heavy tasks—however, our motivation to study them was rooted in the poor performance of MLMs on end-to-end perception reasoning (Fu et al., 2024; Tong et al., 2024b). This hints that MLMs are not optimally utilizing the perceptual capabilities of the visual representations in the language model. We study whether better control of the visual information within the language model could improve MLM perception: specifically, how much perception capability is present in the language model but not currently being surfaced to the final text output?

**Three paired tasks.** Perception-heavy tasks can be framed as visual question answering (VQA); they can also be evaluated directly from the visual representations, i.e., the image value tokens (as in Section 3.1). We refer to these as *paired tasks*. We posit that the image value tokens being able to perform a task on an example is a sufficient (if not necessary, in case the language model can compensate with its reasoning capabilities) condition for the MLM answering the VQA corresponding to the example correctly. We perform this evaluation on three such paired tasks: semantic correspondence, art style and visual similarity, discussed below. We calculate the fraction of instances where the MLM predicts the VQA answer incorrectly, *despite its image value tokens predicting the answer correctly for the same input*.

|  | Semantic Correspondence (%) | Art Style (%) | Visual Similarity (%) |
|---|---|---|---|
| MLM ✓ | 36 | 53.8 | 79.3 |
| Value ✓ MLM ✗ | 7.5 | 33.3 | 18.5 |
| Value ✗ MLM ✓ | 2.5 | 6.0 | 6.7 |
| (MLM ∪ Value) ✓ | **43.5** | **87.1** | **97.8** |

Table 5: Instances in 3 paired tasks with different correctness of the MLM and its image values ("Value"). MLMs under-utilize visual information contained in their image values.

*Semantic correspondence.* We frame semantic correspondence in SPair71K as a VQA task, as in BLINK (for which image coordinates are not public), generating 200 samples. We then evaluate image value tokens on semantic correspondence as an image pair, as in Section 3.1.

*Art style.* We use the Art Style subset of the BLINK benchmark (Fu et al., 2024), which consists of 234 examples that require selecting the image option that shares its art style with the image in the question. We perform average pooling of image value tokens in each image, selecting the image option with highest cosine similarity with the image in the question.

*Visual similarity.* We use the Visual Similarity subset of BLINK (Fu et al., 2024), which consists of 271 examples that require selecting the image option that is most visually similar to the image in the question. We follow the same method as for the art style task.

**Language models significantly under-utilize the visual information they contain.** As shown in Table 5, language models under-utilize visual information present in the model on all four tasks. In the case of the art style task, this is particularly egregious, with performance increasing by 33.3% if the model had successfully surfaced the perception capability of its visual representations. Across all tasks, the percentage of instances where the image value token was incorrect but the MLM prediction was correct is very low (well below random chance, which is 25% for semantic correspondence and 50% for art style and visual similarity), showing the importance of visual representations for these tasks.

Our finding reveals that language models under-utilize their visual information, agreeing with the concurrent Fu et al. (2025) and underscoring the need for further research in better controlling visual information within the language model to improve MLM perception.

## 5 Discussion and Future Work

Our investigation sheds light on an as-yet under-explored component of MLMs: the visual representations stored in the key-value cache and attended to during text generation; in contrast to existing work, which has studied the transformer block outputs within MLMs (Neo et al., 2024; Zhang et al., 2024). We find that: (1) language models refine visual information in earlier layers, then lose it in later layers, with the highest performance being less than that of the equivalent visual encoder that has *not* undergone MLM finetuning on multiple tasks; (2) visual information corresponding to input-agnostic image key tokens in later layers of the language model actually degrades MLM perception; (3) causal attention of language models can be leveraged to improve their visual information via text prefixing; and (4) better control of visual information in the language model could significantly improve MLM perception. We contextualize our findings with relevant work in the Appendix.

Our work calls for research in three main directions: first, improving finetuning of visual encoders like SigLIP in multimodal training so that the visual information provided to the language model after the projection layer is not inferior to that of the original visual encoder without multimodal finetuning; second, improvement of visual representations within the language model by leveraging traditional text-only methods such as prompting; third, improving language model control of existing visual information, potentially by preventing the occurrence of the artifacts we find in later language model layers, or by using additional tokens (Bigverdi et al., 2025). Our work highlights the importance of interpreting image processing in MLMs by studying visual representations in language models, and shows promise for the future of perception reasoning in MLMs.

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

# A Appendix

## A.1 Related work

We discuss three fields of research most relevant to our work.

**Multi-modal language models.** MLMs (Liu et al., 2024; Bai et al., 2023) achieve an end-to-end perception-reasoning system by integrating vision encoders (Radford et al., 2021) with large language models (LLMs) (Chiang et al., 2023; Touvron et al., 2023), leading to significant breakthroughs in various vision-language tasks such as image captioning (Lin et al., 2014) and visual question answering (VQA) (Hudson & Manning, 2019; Goyal et al., 2017). However, recent studies have pointed out that improvements in VQA tasks often stem from the strong language priors of the language model rather than a precise perception of the input image (Fu et al., 2024; Tong et al., 2024b). As a result, MLMs perform poorly on vision-centric benchmarks. Most existing studies on perception focus only on the impact of the vision encoder. In contrast, we investigate how the language decoder influences perception. Concurrent to our work, Fu et al. (2025) reveal that MLMs under-utilize visual information captured by their visual encoders and rely on language priors.

**Mechanistic interpretability.** Although optimizing large language models through next-token prediction has led to many emergent behaviors and significant improvements in various language and vision-language tasks, their internal mechanisms remain largely unknown due to their complexity. This lack of understanding also limits our insights into issues like hallucination. Mechanistic interpretability aims to uncover the internal workings of language models, such as information flow, by using tools like the logit lens (Forum, 2024) and sparse autoencoders (SAEs). However, most existing studies focus on understanding purely language-based models (Templeton et al., 2024; He et al., 2024; Gao et al., 2025). Recently, some works have begun exploring the mechanistic interpretability of multimodal language models (Liu et al., 2025), primarily investigating cross-modal information fusion—specifically, whether the language modality effectively leverages visual information through the attention mechanism (Huo et al., 2024; Jiang et al., 2024; Zhang et al., 2024). However, language models not only involve cross-modal fusion but also process visual information further. While a few studies have explored how language models process visual inputs (Neo et al., 2024), they have not examined the visual representations in the KV cache. Our work focuses on understanding the visual representation stored in the language model and how these representations influence the generated language responses.

**Visual representation learning.** Visual representation learning is one of the core problems in computer vision, where a visual encoder transforms raw pixels into a representation that can be applied to various downstream tasks (Dosovitskiy et al., 2020). The quality of this representation is typically evaluated using probing tasks to assess the model's perception capabilities, on tasks such as ImageNet classification (Deng et al., 2009). In MLMs, commonly used visual encoders include CLIP (Radford et al., 2021) and DINO (Caron et al., 2021). However, in MLMs, the visual encoder is not the only component that processes visual inputs. The cross-modal decoder that follows the encoder not only handles language tokens but also processes visual tokens. Therefore, it is necessary to use probing tasks to analyze the intermediate visual representations produced by the cross-modal decoder.

## A.2 Group Query Attention

In our experiments, we use models that adopt the *Group Query Attention (GQA)* (Ainslie et al., 2023) mechanism. Unlike standard multi-head attention, where each attention head has its own independent query, key, and value projections, GQA shares keys and values across heads while maintaining distinct queries. Specifically, all attention heads in a group share the same key and value matrices but compute attention independently using their own query projections:

$$\text{head}_i = \text{Attention}(Q_i, K_{\text{shared}}, V_{\text{shared}}).$$

This design has two main advantages in our setting. First, most recent state-of-the-art multimodal language models (MLMs) adopt GQA, making our analysis directly applicable

to widely used architectures. Second, by reducing the number of key-value pairs stored in the attention mechanism, GQA greatly simplifies the analysis of attention heads and visual representations, making it more tractable for interpretability research.

In practice, the models we study all adopt this GQA-based architecture. Specifically, both LLaVA-One-Vision-7B and Qwen2.5-VL-7B use 28 queries per layer that share 4 key-value pairs, while LLaMA3-LLaVA-NeXT-8B employs 32 queries per layer sharing 8 key-value pairs.

### A.3 Per-Task Performance of Image Value Tokens Across Layers

We now discuss the performance of image value tokens in LLaVA-OneVision 7B on each of the six probing tasks we evaluate.

On foreground segmentation (Pascal-5i), as shown in Figure 5, the segmentation performance is relatively high in very early layers, then drops gradually over the first 10 layers. It builds again over the next 7 layers, then falls in the remaining layers.

On co-segmentation (MSRC), as shown in Figure 6, the trend in segmentation performance is similar to foreground segmentation: being quite high in early layers, dipping and rising around layer 17, then falling steeply in later model layers.

On semantic segmentation (ImageNet-S), as shown in Figure 7, the segmentation performance continuously builds over the first 18 layers, staying above the performance of the input visual representation (post-projection). It fluctuates slightly in layer layers.

On referring expression segmentation (RefCOCO), as shown in Figure 8, the segmentation performance follows a similar trend as semantic segmentation, building gradually till layer 17 and then dropping gradually, remaining above the input visual representation (post-projection) performance.

On semantic correspondence (SPair71K), as shown in Figure 9, the correspondence performance is about the same as that of the input visual representation (post-projection) initially, then sees a sharp increase in layers 7–10 and sharp decrease in layers 10–13, then a steady decline in the succeeding layers of the model. The variance in image value performance is much greater than that of the segmentation tasks (note the Y axis of the line plot). An extremely similar trend is shown on temporal correspondence (DAVIS), as shown in Figure 10. Both correspondence tasks need to abstract away pixel appearance information, which may explain the steep decline in performance of later language model layers.

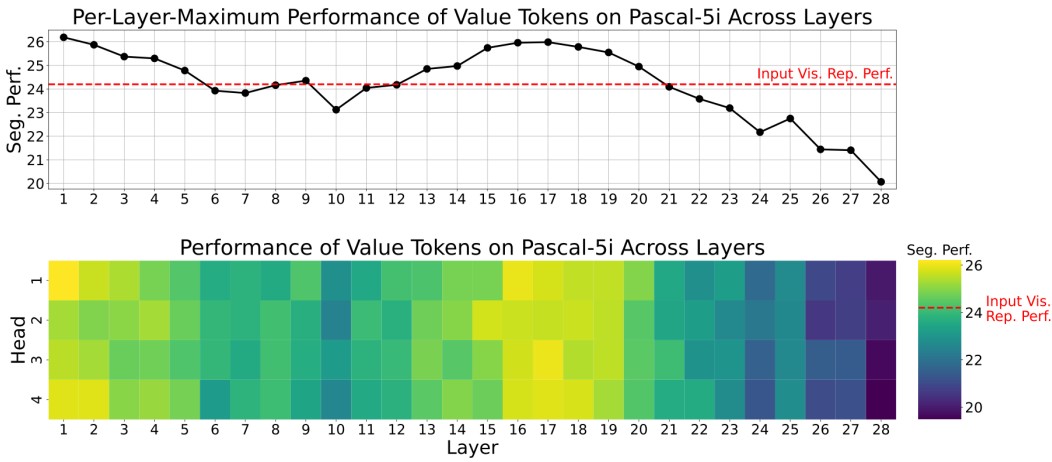

Figure 5: Foreground segmentation performance on Pascal-5i for LLaVA-OneVision 7B. (top) Maximum-per-layer performance. (bottom) Performance per head.

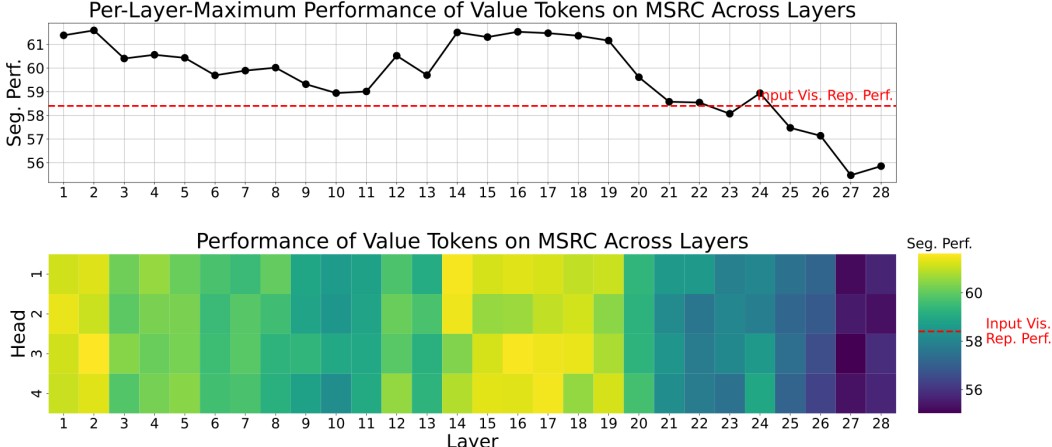

Figure 6: Co-segmentation performance on MSRC for LLaVA-OneVision 7B. (top) Maximum-per-layer performance. (bottom) Performance per head.

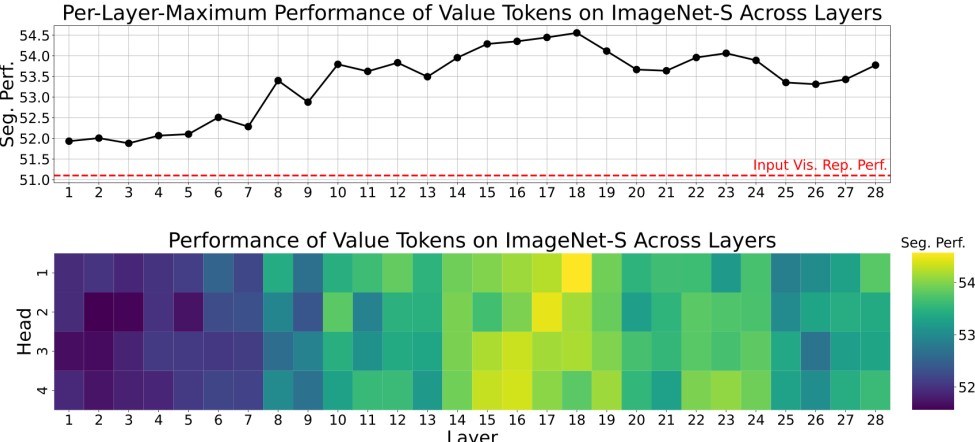

Figure 7: Semantic segmentation performance on ImageNet-S for LLaVA-OneVision 7B. (top) Maximum-per-layer performance. (bottom) Performance per head.

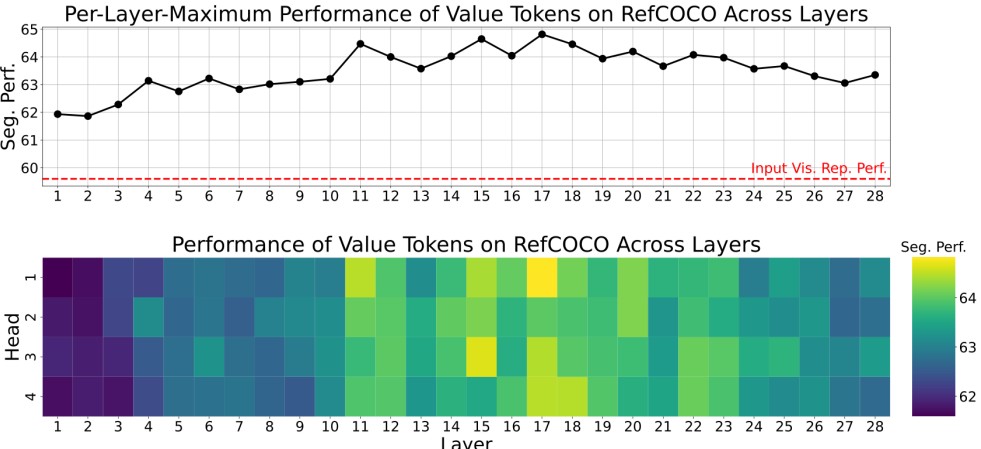

Figure 8: Referring expression segmentation performance on RefCOCO for LLaVA-OneVision 7B. (top) Maximum-per-layer performance. (bottom) Performance per head.

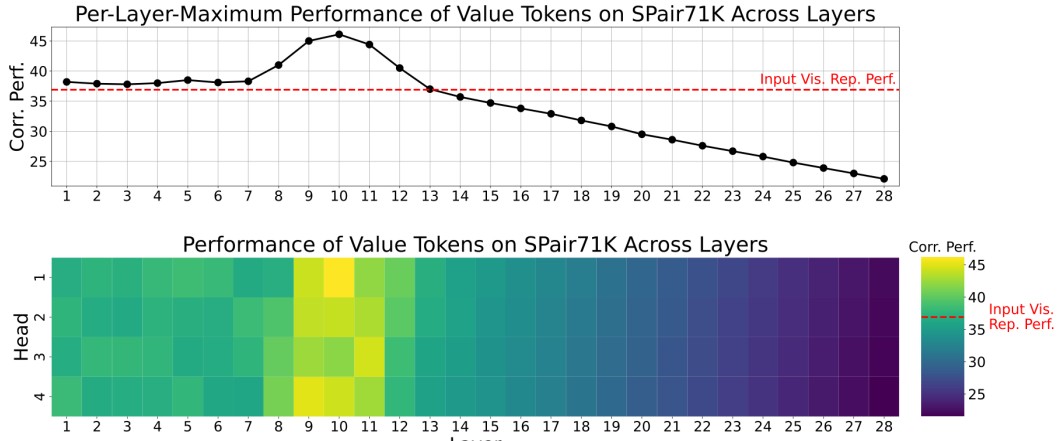

Figure 9: Correspondence performance on SPair71K for LLaVA-OneVision 7B. (top) Maximum-per-layer performance. (bottom) Performance per head.

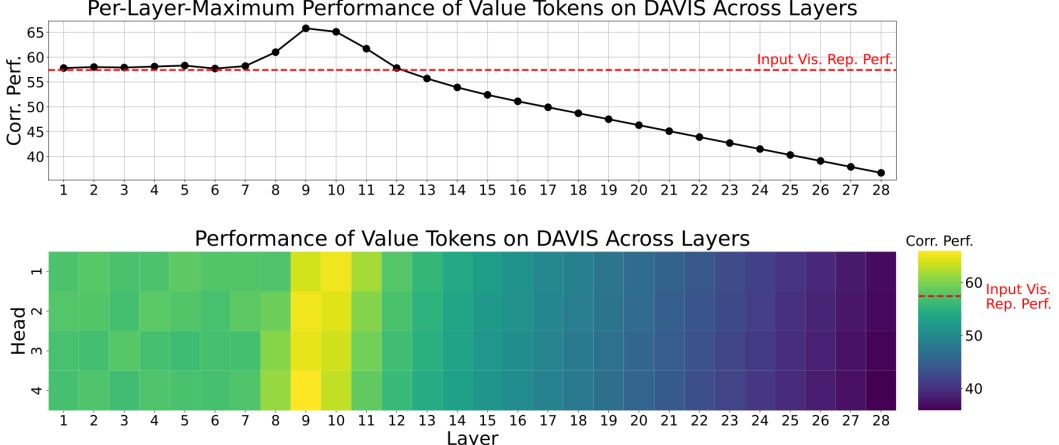

Figure 10: Correspondence performance on DAVIS for LLaVA-OneVision 7B. (top) Maximum-per-layer performance. (bottom) Performance per head.

### A.4 Segmentation Results for All Models

We report segmentation results of all three MLMs in Table 6. Qwen2.5-VL 7B performs the best, closely followed by LLaVA-OneVision 7B. Llama3-LLaVA-NeXT 8B performs less well, as its vision and text components both exhibit lower performance than those of the other two models.

We also show the performance of image value tokens averaged across the four segmentation tasks of each of these models: LLaVA-OneVision 7B (Figure 11), Qwen2.5-VL 7B (Figure 12), and Llama-3-LLaVA-NeXT 8B (Figure 13). All three models exhibit a similar trend: the performance of their value tokens on segmentation increases gradually over the first two-third layers, then drops steeply in the later layers.

| Visual representations | Pascal-5i | MSRC | ImageNet-S | RefCOCO | Average |
|---|---|---|---|---|---|
| LLAMA3-LLaVA-NeXT 8B | 21.6 | 54.4 | 51.4 | 62.6 | 47.5 |
| LLaVA-OneVision 7B | 26.2 | 61.6 | 54.6 | 64.8 | 51.8 |
| Qwen-2.5VL 7B | **28.5** | **63.9** | **55.9** | **65.8** | **53.5** |

Table 6: Segmentation results of the best value in MLMs, split across 4 tasks. Results are in mIOU but for MSRC co-segmentation, which is reported in $\mathcal{J}_m$.

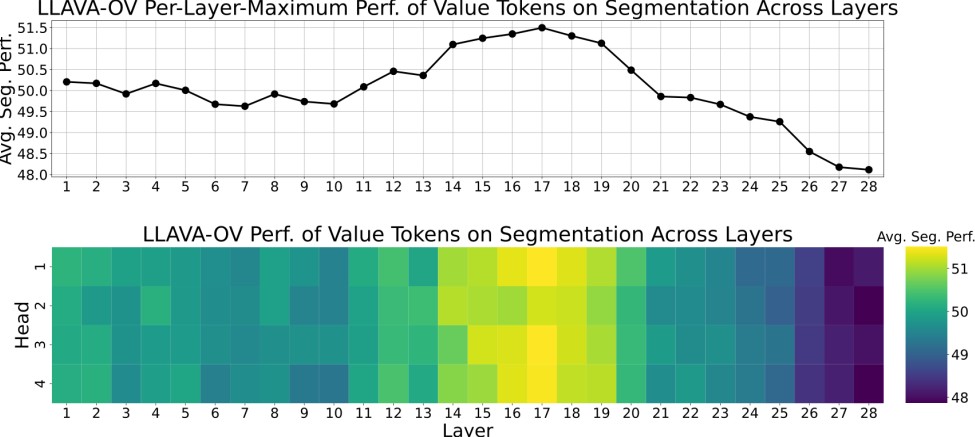

Figure 11: Average segmentation across 4 tasks for LLaVA-OneVision 7B. (top) Maximum-per-layer performance. (bottom) Performance per head.

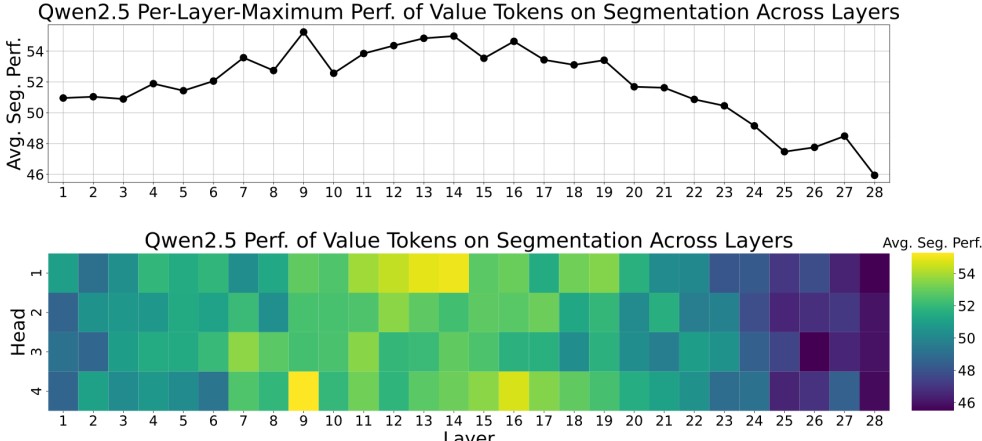

Figure 12: Average segmentation across 4 tasks for Qwen2.5-VL. (top) Maximum-per-layer performance. (bottom)

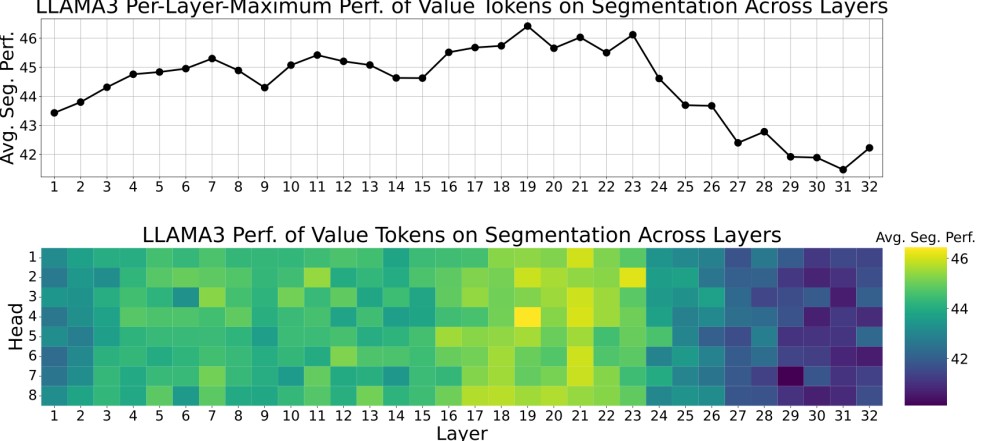

Figure 13: Average segmentation across 4 tasks for LLaMA-3-LLaVA-NeXT. (top) Maximum-per-layer performance. (bottom) Performance per head.

### A.5 Correlation of Perception Capability of Image Value Tokens with that of the overall MLM: Additional Details

We assess MLMs' end-to-end perception reasoning by evaluating them on the perception-heavy vision-language reasoning tasks in CV-Bench (Tong et al., 2024a) and BLINK (Fu et al., 2024). This allows us to investigate a key question: *Does improved perception of visual representations in the language model correlate with better perception reasoning of the MLM itself?* If so, this would motivate the need to further study visual representations within the language model.

**Experiment.** To calculate MLMs' end-to-end perception reasoning capability, we evaluate on two vision-centric VQA benchmarks: CV-Bench, and the perception-heavy tasks in BLINK. CV-Bench consists of 2,637 test samples and assesses MLMs' 2D and 3D perception capabilities in a single image, covering four major categories: counting, relations, depth estimation, and distance estimation. BLINK, on the other hand, contains single-image and multi-image questions. Since certain question types in BLINK, such as art style, are less relevant to perception, we select a subset of BLINK consisting of four perception-related tasks: counting, object localization, relative depth, and spatial relations, totaling 1,021 test samples. We evaluate MLMs across these 3,658 test samples to estimate their perception reasoning ability. We compare this to MLM image value tokens' segmentation capabilities averaged across our four segmentation probing tasks: foreground segmentation, co-segmentation, semantic segmentation and referring expression segmentation.

We evaluate two MLMs with different configurations: (1) models of the same version but different sizes: LLaVA-OneVision (0.5B, 7B, 72B); and (2) models of the same size but different versions: Qwen2-VL 7B and Qwen2.5-VL 7B. The LLaVA-OneVision 0.5B model stores 2 key-value pairs per layer, the 7B model stores 4 key-value pairs per layer, and the 72B model stores 8 key-value pairs per layer.

**Results.** As shown in Figure 3 and discussed in Section 3.1, we observe that MLMs of different sizes and versions exhibit the same trend: the stronger the perception capability of their image value tokens, as estimated by their segmentation performance, the better the MLMs themselves perform on downstream perception reasoning vision-language tasks. This finding shows that improved perception of visual representations in the language model goes hand-in-hand with better perception reasoning capabilities of the MLM itself on perception-heavy vision-language tasks—showing the importance of high-granularity visual representations, and further emphasizing the need to study the same in MLMs.

### A.6 Input-Agnostic Visual Representations: Additional Details

#### A.6.1 Identifying Input-Agnostic Image Key Tokens

As discussed in Section 3.2, we study the variance in image keys per head per layer across 1000 COCO images. The variance is shown in Figure 14 as a distribution with the threshold depicted and in Figure 15 as a heatmap.

We find that when plotting the variance of each key, a bimodal distribution forms, with the first mode centered at variance of 200, and the second mode centered at variance of 600. We set the threshold at the lowest point between these two modes, at a variance of 450. Keys with variance below the threshold are labeled input-independent, and are labeled input-dependent otherwise. Note: to ensure our experiments are not specific to a single image distribution, we calculated variance across validation set of ADE20K (which consists of 2000 images). The pattern in variance was consistent.

We find that when setting this threshold: 20 keys are input-independent in the first 8 layers (including all 16 keys in the first four layers); 8 keys are input-independent in the next 10 layers; and 14 keys are input-independent in the last 10 layers. We call these the early layers, middle layers and late layers respectively.

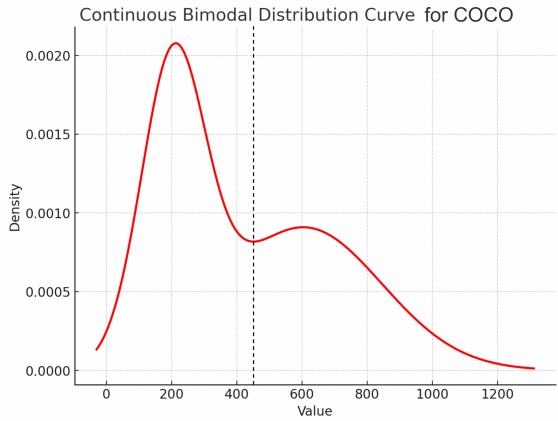 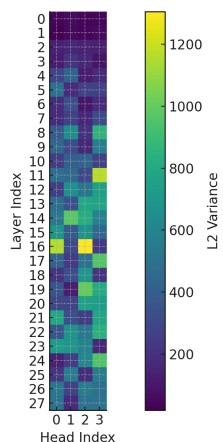

Figure 14: Distribution of L2 variance of image keys in LLaVA-OneVision 7B across 1000 COCO images. We set a threshold at variance=450 (vertical dashed line) to determine whether keys are input-agnostic or input-dependent.

Figure 15: Heatmap depicting the L2 variance of each image key token in LLaVA-OneVision 7B across 1000 COCO images.

### A.6.2   Intervention on Input-Agnostic Image Key Tokens

We then run the intervention by blocking: (1) only the input-independent keys at early layers; (2) only the input-independent keys at middle layers; and (3) only the input-independent keys at late layers.

We perform this experiment for both LLaVA-OneVision 7B and Qwen2.5-VL 7B. We show results on POPE in Table 7. We also show results on another popular hallucination benchmark: MME. Following the setup from Zhang et al. (2025), we evaluate on four subsets of MME: count, color, position and existence. In all four of these, the model is asked a Yes/No question about the image (similar to POPE), which address hallucination across various axes and also measure general model capabilities. Results on MME are shown in Table 8.

Our results clearly show the applicability of our findings to multiple benchmarks and multiple models: specifically, the existence of input-agnostic image key tokens in later layers of the model, which (similar to the finding from Darcet et al. (2024)) encode artifacts. Note: the "existence" subset, which is the closest equivalent to POPE, is much smaller than POPE (60 data points compared to 9000 in POPE). The models seem to achieve perfect existence scores both before and after intervening, hinting that this existence subset is both smaller and easier than POPE.

| Model | Intervention | POPE F1 |
|---|---|---|
| LLaVA-OV | none (base model) | 78.1 |
| | early layers | 65.1 |
| | middle layers | 77.9 |
| | late layers | **81.2** |
| Qwen2.5-VL | none (base model) | 87.9 |
| | early layers | 69.1 |
| | middle layers | 82.4 |
| | late layers | **89.2** |

Table 7: Intervention study on LLaVA-OneVision 7B and Qwen2.5-VL 7B on the POPE benchmark. Intervening at middle-to-late layers mitigates halllucination.

| Model | MME Subset | Score of Base Model | Score after Intervention |
|---|---|---|---|
| LLaVA-OV | count | 170.0 | **175.0** |
| | color | 178.3 | **187.5** |
| | position | 145.0 | **153.3** |
| | existence | **200.0** | **200.0** |
| Qwen2.5-VL | count | **182.5** | **182.5** |
| | color | 185.8 | **193.3** |
| | position | 160.0 | **165.8** |
| | existence | **200.0** | **200.0** |

Table 8: Intervention study on LLaVA-OneVision 7B and Qwen2.5-VL 7B on the MME benchmark. Intervening at middle-to-late layers mitigates hallucination and improves general model capability.

### A.6.3 Controls for the Intervention Experiment

To ensure that artifacts are present in input-agnostic representations alone, we additionally perform this intervention on: (1) the same number of input-dependent keys in later layers, randomly selected; and (2) the same number of all keys in later layers, randomly selected.

Results from controls on POPE are shown in Table 9, and results from controls on MME are shown in Table 10. In all cases, the controls under-perform the main experiment, showing that only representations corresponding to input-agnostic keys contain artifacts.

| Model | Intervention | POPE F1 |
|---|---|---|
| LLaVA-OV | none | 78.1 |
| | input-agnostic | **81.2** |
| | input-dependent | 77.4 |
| | random | 78.6 |

Table 9: Control for the intervention study on LLaVA-OneVision 7B on POPE. In each experiment, the same number of keys were blocked in the same layers of the model. Blocking input-agnostic keys improves performance, while blocking input-dependent keys reduces it. Blocking a random combination of them performs midway between the two. Clearly, only representations corresponding to input-agnostic keys contain artifacts.

| Model | Intervention | MME subset | | | |
|---|---|---|---|---|---|
| | | count | color | position | existence |
| LLaVA-OV | none | 170.0 | 178.3 | 145.0 | **200.0** |
| | input-agnostic | **175.0** | **187.5** | **153.3** | **200.0** |
| | input-dependent | 165.8 | 172.5 | 141.7 | 197.5 |
| | random | 169.2 | 175.0 | 146.7 | 199.2 |

Table 10: Control for the intervention study on LLaVA-OneVision 7B on MME. In each experiment, the same number of keys were blocked in the same layers of the model. Similar to POPE, blocking input-agnostic keys improves performance, while blocking input-dependent keys reduces it. Blocking a random combination of them performs midway between the two. Clearly, only representations corresponding to input-agnostic keys contain artifacts.

## A.7 Image Key Token Visualizations

Figures 16, 17 and 18 show key visualizations for samples from COCO for LLaVA-OneVision 7B, Qwen2.5-VL 7B and Llama3-LLaVA-NeXT 8B respectively.

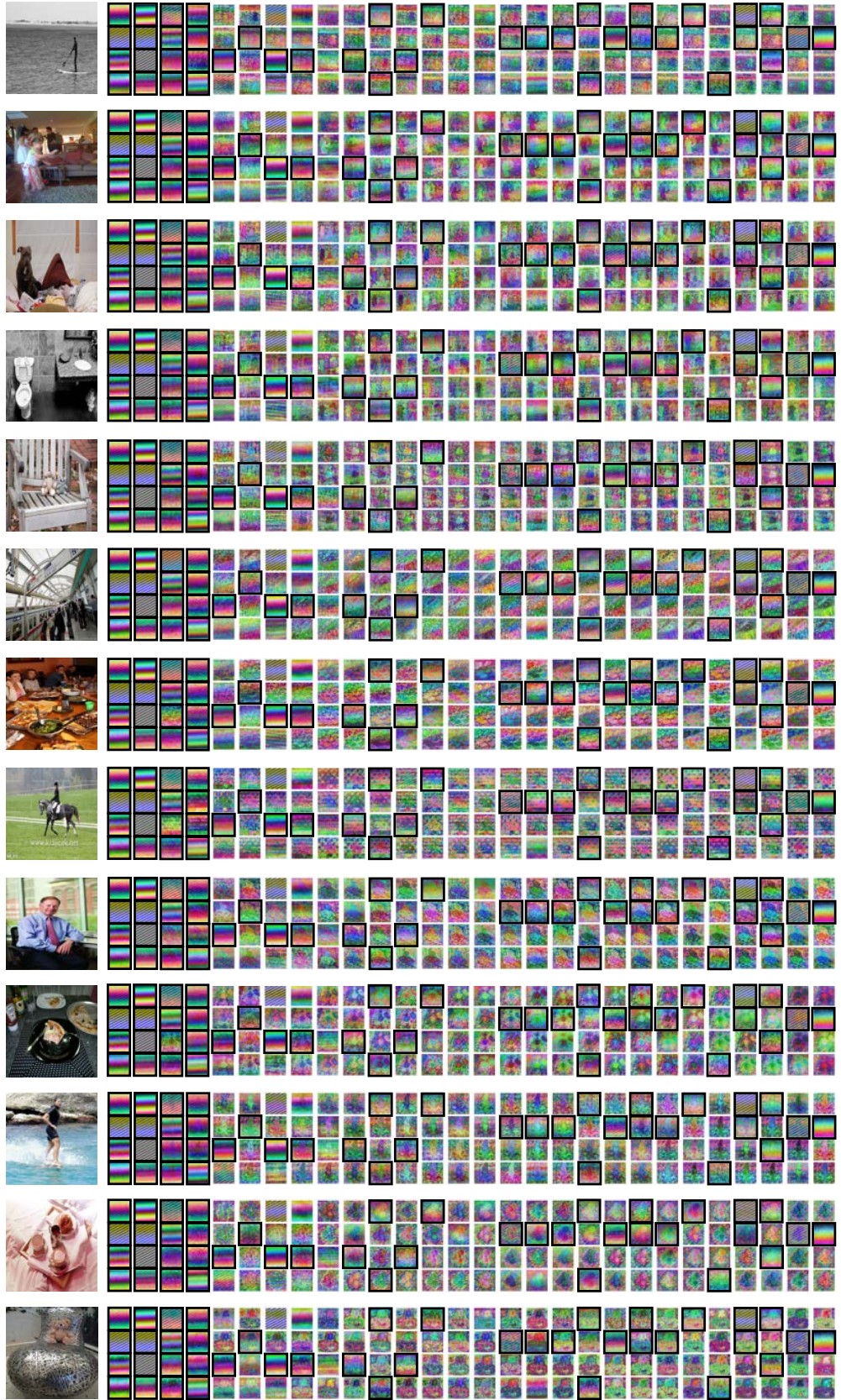

Figure 16: Image keys in LLaVA-OneVision 7B for a random sample of COCO images, with input-agnostic keys highlighted.

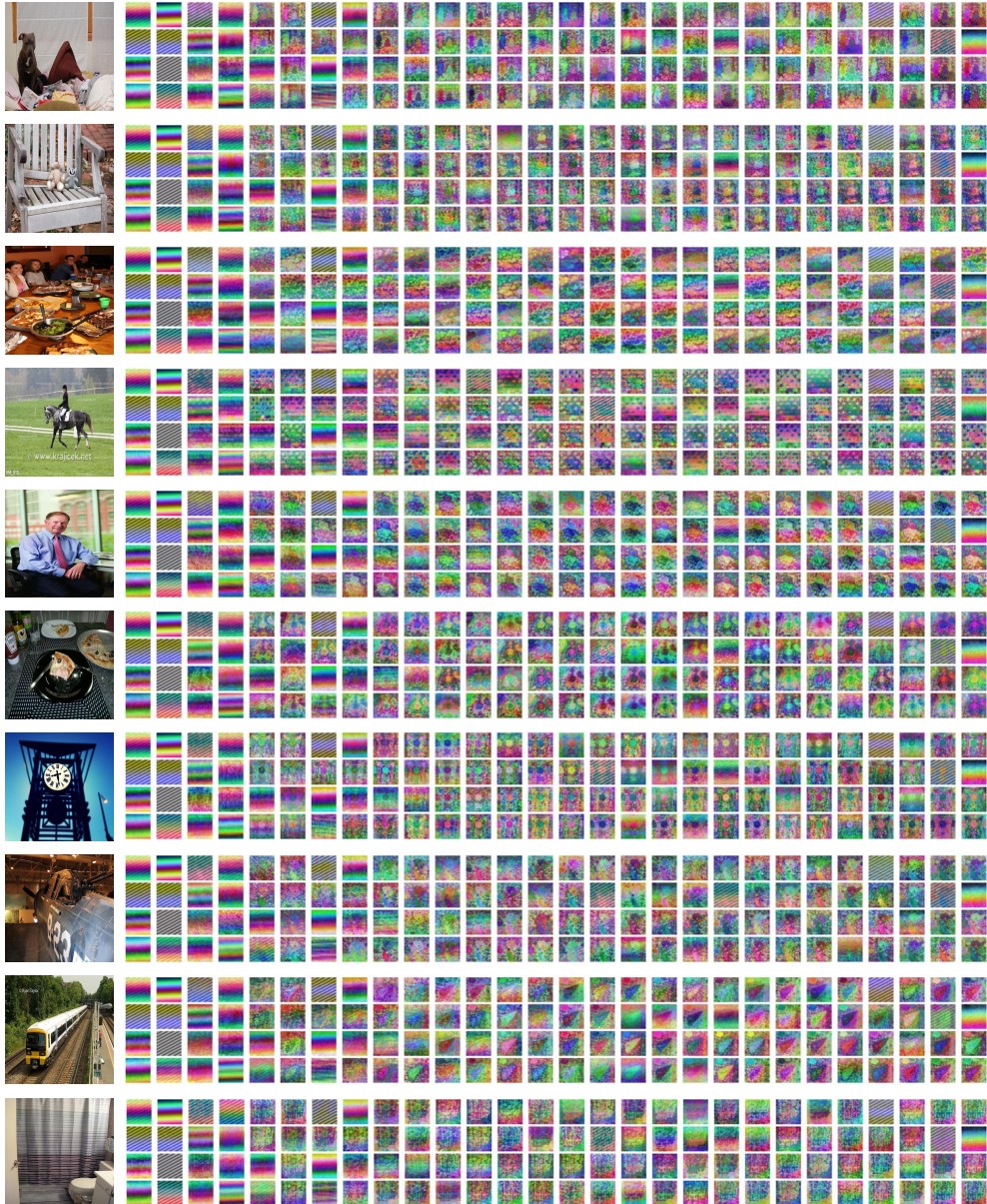

Figure 17: Image keys in Qwen2.5-VL 7B for a random sample of COCO images.

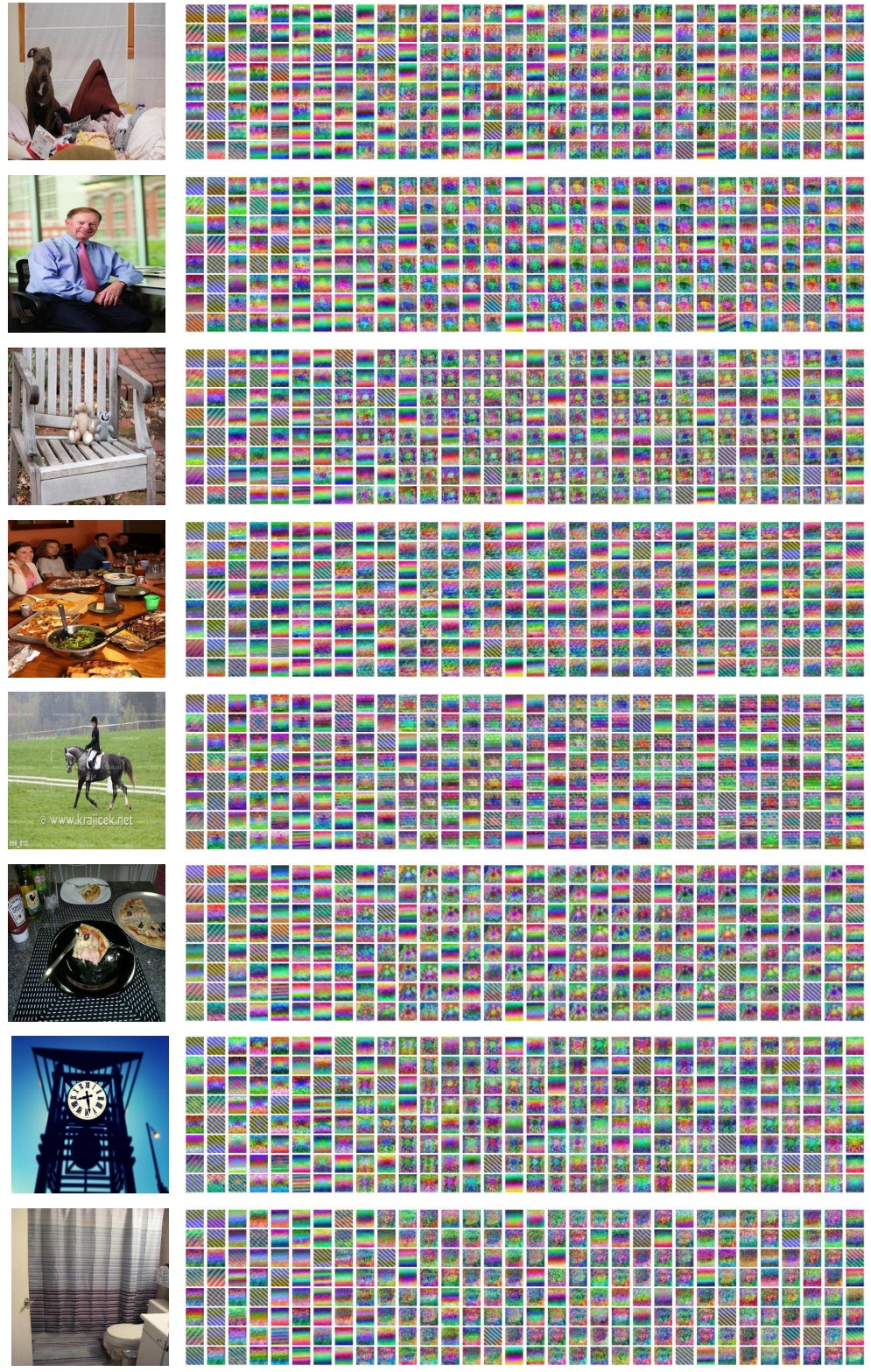

Figure 18: Image keys in Llama-LLaVA-NeXT 8B for a random sample of COCO images. This model stores eight KV pairs per layer. Their quality is less than that of LLaVA-OneVision 7B, as their vision and text components are both weaker.

