# OpenReview forum: "Visual Representations inside the Language Model"
_colmweb.org/COLM/2025/Conference — COLM 2025_

### Official Review · Reviewer_6is2 · 2025-05-12

**Rating:** 7
**Confidence:** 4
**Ethics Flag:** 1

**Summary:**

The paper presents the results of a number of analyses focusing on the visual key-value store of three multimodal language models (MLM).
The paper identifies keys which do not depend on the visual input (i.e. they are nearly constant across input images) and shows that by blocking attention to these keys in upper layers of MLMs can reduce "hallucinations".
The next analysis shows that visual value tokens encode information sufficient for several vision task, and correlated with model performance on these downstream tasks. However, this information is not always successfully surfaced by the model to correctly perform the downstream task.

**Questions To Authors:**

You may want to be explicit about what exactly  the key visualizations depict, i.e. what does each pixel in the "feature map" stand for.
Table 3 contains some symbols and terms which are not referred to in the caption or the text (I_m, F_m, αbbox).
It is not clear how the numbers shows in figure 3 are derives from the performance scores on the tasks described in the text.
The paragraph in section 2 named "Mechanistic interpretability" would likely be better named just interpretability as there is little in covered in there that's particularly mechanistic.

**Reasons To Accept:**

The paper is focused study of a specific, little studied aspect of MLM behavior. The analyses are appropriate for the task and carried out properly. The results are of interest to the vsion and language field and may prove useful to improve the understanding and performance of MLMs.

**Reasons To Reject:**

The quality of presentation in some places could be improved. The details of several tables and figures are hard to interpret. The contribution of the paper is very focused and therefore somewhat narrow.

---

> ### Author Response · Authors · 2025-06-03
> **Response to Reviewer #4**
>
> We thank Reviewer 4 for your time and detailed comments. We address them below.
>
> Thank you for your feedback about the presentation of our paper. We will improve the writing and clarify details about the tables and figures, including the metrics defined in Table 3, the feature maps, and Figure 3.
>
> Regarding our contribution, we have added further results in General Response to Reviewers, expanding significantly on our experiments in the paper on intervention to prevent model hallucination. Our contributions thus tackle a wide range of tasks related to perception capability of MLMs: *hallucination detection, few-shot foreground segmentation, co-segmentation, semantic segmentation, referring expression segmentation, semantic correspondence and temporal correspondence*. We would be happy to provide further results during the discussion period, per suggestions you may have.
>
> Thank you again, and please let us know if you have any questions, or suggestions for further experiments. We will be happy to answer and provide the same.

---

> ### Comment · Area_Chair_kWGv · 2025-06-07
> **Discussion**
>
> Dear Reviewer 6is2,
>
> The authors have responded to your review. Does their answer address your questions and reasons to reject?

---

> > ### Comment · Reviewer_6is2 · 2025-06-10
> >
> > Thanks to the authors for their response, I appreciate them elaborating on the applicability of the method. I will keep my rating as it is already quite favorable to the paper.

---

> ### Author Response · Authors · 2025-06-09
> **Are there any further questions/concerns we could address?**
>
> We once again thank all the reviewers for their time. We believe we have addressed all questions and concerns in our response, and that our paper will certainly be stronger upon incorporating these new results.
>
> We wanted to follow up in advance of the upcoming deadline — we hope our response may lead you to consider updating your rating of our work.
>
> We have summarized our response and our contributions in "General Response #2" above.
>
> Please let us know if you have any remaining questions or concerns that we could clarify!

---

### Official Review · Reviewer_GABW · 2025-05-12

**Rating:** 6
**Confidence:** 4
**Ethics Flag:** 2

**Summary:**

This paper delves into the visual representations within multimodal language models (MLMs), focusing on the visual key-value (KV) tokens in the LLM. Through experiments on three popular MLMs, the authors differentiate between input-dependent and input-independent keys, revealing that blocking attention to input-independent keys in middle-to-later layers can reduce hallucinations in object identification tasks by 3.2%.  Furthermore, they demonstrate that value tokens encode sufficient information for perception-intensive tasks such as segmentation, semantic correspondence, and temporal correspondence. These value tokens even outperform strong ViT-based models like CLIP and DINO on certain tasks. The study underscores the potential of KV tokens for improving perception capabilities in MLMs and highlights the importance of cross-modality for vision-only tasks. The paper's contributions are significant, offering new insights into the interpretability of MLMs and providing a novel perspective for enhancing their perception performance.

**Questions To Authors:**

See 'Reasons To Reject' section above

**Reasons To Accept:**

1. Innovative Research Perspective: The paper focuses on the KV tokens within LLMs, a relatively underexplored area. By analyzing these tokens, the authors provide a fresh perspective on understanding how MLMs process visual information, distinguishing their work from prior studies that primarily focused on vision encoders or transformer block outputs.
2. Strong Empirical Evaluation: The paper includes extensive experiments on three different MLMs, covering tasks such as segmentation, semantic correspondence, and temporal correspondence. The results demonstrate that value tokens possess substantial perceptual capabilities, providing solid empirical support for the claims.
3. Practical Significance: The findings reveal that cross-modality can benefit vision-only tasks, offering valuable insights for improving perception performance in MLMs. For instance, the paper shows that prefixing textual prompts to images can enhance the performance of value tokens on tasks like semantic segmentation, highlighting the potential for leveraging cross-modality to boost perception capabilities.

**Reasons To Reject:**

The authors categorize visual key tokens into input-dependent and input-independent keys based on whether the key variance across 1K COCO images exceeds a fixed threshold. The following issues may limit the generalizability of the findings:
1.  How does blocking attention to input-independent keys in middle-to-later layers affect POPE under different threshold settings?
2. As all input images are from COCO, do input-independent keys relate to image distribution?
3. Given that the intervention studies on image keys are conducted on a relatively small scale, do input-independent keys remain consistent when scaled up?
4. Will blocking input-independent keys improve performance on other benchmarks beyond the illusion benchmark? We suggest that the authors also evaluate their approach on general-purpose benchmarks like MMBench.

---

> ### Author Response · Authors · 2025-06-03
> **Response to Reviewer #3**
>
> We thank Reviewer #3 for your time and detailed comments. We address them below.
>
> Thank you for your suggestion regarding further experiments in our hallucination study. We have conducted further experiments with hallucination intervention, which are presented in the General Response to Reviewers. We will provide further results during the discussion period as well, such as your suggestion to evaluate on other benchmarks beyond POPE.
>
> Thank you again, and please let us know if you have any questions, or suggestions for further experiments beyond these. We will be happy to answer and provide the same.

---

> ### Comment · Area_Chair_kWGv · 2025-06-07
> **Discussion**
>
> Dear Reviewer GABW,
>
> The authors have responded to your questions. Does their answer address your questions and reasons to reject?

---

> ### Author Response · Authors · 2025-06-09
> **Are there any further questions/concerns we could address?**
>
> We once again thank all the reviewers for their time. We believe we have addressed all questions and concerns in our response, and that our paper will certainly be stronger upon incorporating these new results.
>
> We wanted to follow up in advance of the upcoming deadline — we hope our response may lead you to consider updating your rating of our work.
>
> We have summarized our response and our contributions in "General Response #2" above.
>
> Please let us know if you have any remaining questions or concerns that we could clarify!

---

> ### Comment · Reviewer_GABW · 2025-06-09
> **Reviewer Decision**
>
> The author's responses indeed addressed some of my concerns, it would be beneficial to see more evaluations on general-purpose benchmarks to better assess the practical applicability of the proposed approach; however, this was not provided. Having considered the other reviewers' comments and the author's response, from my point of view, this paper indeed provides a good insight for visual representations for VLM research. Therefore, my initial assessment remains unchanged.

---

> > ### Author Response · Authors · 2025-06-10
> > **Further results on general-purpose benchmarks**
> >
> > We thank Reviewer #3 for their response. Per your suggestion, we have run further experiments in our hallucination intervention setting (from Section 4.2 of the paper): beyond POPE, we have also evaluated on multiple subsets of MME: count, color, position and existence.  Our results (as shown in General Response #3) show that our interventions ensure that the model improves (or at least retains) performance: across multiple models, and multiple benchmarks.
> >
> > Our results from all other experiments (Section 5.1-5.5) are also run across multiple tasks: few-shot foreground segmentation, co-segmentation, semantic segmentation, referring expression segmentation, semantic correspondence and temporal correspondence, and reveal the importance of further research in how language model layers process visual tokens, in order to improve the perception capabilities of VLMs. (These experiments probe/interpret the model without intervening, and thus retain the base model performance on general-purpose benchmarks).
> >
> > We believe we have addressed your concern about general-purpose performance, but please let us know if there is anything further we could clarify. We look forward to hearing from you!

---

### Official Review · Reviewer_SMbp · 2025-05-12

**Rating:** 5
**Confidence:** 5
**Ethics Flag:** 1

**Summary:**

* The work addressed the issue of Multimodal Language Models (MLMs) performing poorly on perception-heavy tasks.

* Instead of improving the architecture, the paper proposes investigating key-value (KV) tokens to understand how MLMs process visual information.

* The work first distinguishes between input-dependent and input-independent keys and finds that it correlates with specific hallucination issues. Then, the paper investigates the image value tokens that perform descent on image-only tasks.

* The findings offer more discussion into MLM functioning, which aims to acquire deeper mechanistic interpretability.

**Questions To Authors:**

* In section 5.4, I don't get what the point the author is trying to make. Could you elaborate more on this?

**Reasons To Accept:**

* The paper is well written and easy to follow.

* The preliminaries in section 3.1 help understand the paper's following claims. Also, it provides a good comparison between different architectures and their purposes.

* Visualization helped understand the low variance claim.

**Reasons To Reject:**

This work tries to lead a discussion with several findings. However, most of the claims lack experimental support.

1. In Section 4.1, the work tried to bound the connection between feature variance and hallucination. However, the motivation is unclear, and it is not apparent that they are related. In addition, the experiment of blocking independent keys is insufficient. Since the paper investigated three models, why not show the results of the three? Also, it was performed on a single benchmark. It is hard to claim the independent key is the actual cause of hallucination.

2. In section 5.1, the paper tried to make a claim that MLM can perform on image tasks. Table 1 should be compared with existing segmentation models to show how close the models can achieve. For the prefixing setting, it is evident that writing a context-related prompt may improve MLM performance, which is the reason why MLMs were developed. The new insight is limited.

3. Same for section 5.2, the finding is the motivation of MLMs

4. In section 5.3, the work tried to compare MLM with CLIP and DINO, which is not a fair comparison considering different training data and model architectures.

---

> ### Author Response · Authors · 2025-06-03
> **Response to Reviewer #2**
>
> We thank Reviewer 2 for your time and detailed comments. We address them below.
>
> - **Reasons to Reject #1: Connection between feature variance and hallucination + Hallucination intervention only done on one model and benchmark.**
>   - *Connection between feature variance and hallucination:* We draw inspiration from Darcet et al “Vision Transformers Need Registers” (CVPR 2024), which showed that middle layers of the transformer corresponding to background patches encode input-agnostic artifacts. Combined with our insight that several of the image key tokens were input-agnostic, we wanted to determine whether these, too, encoded input-agnostic artifacts. We chose a task that requires careful attention paid to the input: hallucination detection.
>
>     We found that input-independent keys in the initial layers, while input-agnostic, encode important information (as removing them significantly reduced model performance), while input-independent keys in later layers encode artifacts (as blocking them mitigated hallucination). We will expand on this discussion in the paper.
>   - *Hallucination intervention only done on one model and one benchmark:* Thank you for your suggestion. We have conducted further experiments with hallucination intervention, which are presented in the General Response to Reviewers. We will provide further results during the discussion period as well.
>
> - **Reasons to Reject #2: Table 1 should be compared with existing segmentation models to show how close the models can achieve.**
>   - Thank you for your suggestion. Segmentation models achieve much higher results on segmentation, having been designed and trained specifically for the task. Our goal here is to evaluate MLM’s *intermediate representations* on perceptual tasks such as segmentation (to study further MLMs’ poor performance on perception). While not close to state-of-the-art (e.g., state-of-the-art on ImageNet-S is 63.2 MIOU, whereas LLaVA-OneVision 7B intermediate representations achieve 54.6 MIOU, per Table 4), the results are strong enough to show that MLMs should be doing better on perception tasks, given their intermediate representations’ performance (as discussed in Section 5.5). We will add these numbers and a discussion to the paper.
>
> - **Reasons to Reject #3: For the prefixing setting, it is evident that writing a context-related prompt may improve MLM performance, which is the reason why MLMs were developed. The new insight is limited.**
>   - In this experiment, we are not evaluating the MLM overall, but its intermediate representations: specifically, the image value tokens from the MLM layers’ key-value cache.
>   - It is also worth noting that almost all existing evaluations of MLMs (e.g., LLaVA) place the image first, followed by the text. We here show that perception capabilities (in the context of domain shift) of the intermediate representations can actually improve if this order is reversed.
>
> - **Reasons to Reject #4: In section 5.3, the work tried to compare MLM with CLIP and DINO, which is not a fair comparison considering different training data and model architectures.**
>   - In this experiment as well, we are not evaluating the MLM overall, but its intermediate representations (image value tokens from the MLM layers’ key-value cache). It is not a direct comparison to CLIP and DINO, but rather a point of reference to highlight the capabilities of the intermediate representations. We will clarify this in the writing, thank you for highlighting this point.
>
> -------
>
> - **Questions to Authors #1: In section 5.4, I don't get what the point the author is trying to make. Could you elaborate more on this?**
>   - Figure 3 shows the correlation between performance of the MLM’s intermediate representations on perception tasks (X-axis) and the performance of the MLM itself on perception tasks (Y-axis). We see a strong correlation between the two, highlighting the importance of studying intermediate representations of MLMs.
>    - Note: To isolate the number of parameters, we also evaluate two models with the same number of parameters (Qwen2-7B and  Qwen2.5-7B).
>
>
> Thank you again, and please let us know if you have any questions, or suggestions for further experiments. We will be happy to answer and provide the same.

---

> > ### Comment · Reviewer_SMbp · 2025-06-09
> >
> > In Reasons to Reject #3, the author stated, "We here show that perception capabilities (in the context of domain shift) of the intermediate representations can actually improve if this order is reversed."
> >
> > At which part of the paper is showing this claim and why? I didn't follow this.

---

> > > ### Author Response · Authors · 2025-06-10
> > >
> > > Thank you for your question: we refer to the experiments in Section 5.2, where we show that the visual tokens within the language model can adapt to a text prefix and improve on the task at hand, even on perception-heavy tasks like semantic segmentation.
> > >
> > > In VLM evaluations, the image is usually put before the text. This works in general chatbot-like settings, where one doesn't want the image representations to depend too heavily on the first turn of text (which is likely to be a specific question about the image), because they may become worse at responding to later turns.
> > >
> > > However, we show that there are some cases in which this can actually be an advantage. In our experiment in Section 5.2, we show that by prefixing the name of the domain before the image, the visual representations within the language model adapt well to the domain, showing improvements even on perception-heavy tasks like semantic segmentation (as well as referring expression segmentation).

---

> ### Comment · Area_Chair_kWGv · 2025-06-07
> **Discussion**
>
> Dear Reviewer SMbp,
>
> The authors have responded to your questions. Does their answer address your questions and reasons to reject?

---

> ### Author Response · Authors · 2025-06-09
> **Are there any further questions/concerns we could address?**
>
> We once again thank all the reviewers for their time. We believe we have addressed all questions and concerns in our response, and that our paper will certainly be stronger upon incorporating these new results.
>
> We wanted to follow up in advance of the upcoming deadline — we hope our response may lead you to consider updating your rating of our work.
>
> We have summarized our response and our contributions in "General Response #2" above.
>
> Please let us know if you have any remaining questions or concerns that we could clarify!

---

> > ### Comment · Reviewer_SMbp · 2025-06-09
> >
> > Thanks to the author for the response. Do you have any further results regarding Reason to Reject #1?

---

### Official Review · Reviewer_RSnv · 2025-05-19

**Rating:** 4
**Confidence:** 4
**Ethics Flag:** 1

**Summary:**

The motivation for this paper is that we don’t know why MLMs struggle with semantic segmentation and object localization.  Previous work looked at ViTs or the Transformer blocks.   KV tokens are less studied here.  These are studied across LLaVA-OneVision and other models and input independent and dependent keys are identified.  It is found that value tokens contain enough information for these tasks and outperform clip and Dino.  Notably perception information doesn’t necessarily make it through to the final output, even when the information is present.  It is found that blocking text queries to image keys mitigates hallucination in the POPE object identification task by 3.2%.  Section 2 describes related work.  Section 3 describes the preliminaries, including MLMs and an image encoder, lightweight adapter and language decoder.  Three key differences in the image processing in the language model are 1) explicit key-value store 2) causal attention and 3) cross-attention between modalities.  Input independent keys have very low variance across input images while input dependent ones have much higher variance.  These tend to dominate mid-to-later layers of the model.  Each question in the POPE benchmark asks whether a particular object appears in the image (9000 examples).

An intervention is performed to block text tokens on input independent keys in 1) the first four layers and then 2) the last 10 layers.  Section 5 describes image values.  The authors hypothesize that image values contain well localized semantics while maintaining cross-modal alignment: segmentation and correspondence are looked at.  All tasks are zero-shot.  Few shot segmentation, semantic segmentation, referring expression segmentation and semantic and temporal correspondence are all evaluated.

Section 5.2 indicates that visual representations are promotable by prefixing language.  Prefixing improves performance.  Section 5.3 shows mixed results though the authors are investigating the power of the cross modal representations.  They finally evaluate things on reasoning tasks.  The better the values on image tasks (segmentation), the better on the final output.  Conclusion is that MLMs should be doing better at perception than they are.  The authors conclude with future work.

**Questions To Authors:**

* Can you please clarify the way that you think about the interaction between visual encoders (SigLip, CLIP, DINO) and MLMMs?  And how that factors into the results?  I would expect at least a discussion in the paper on this.  MLMMs should be strictly better.
* What does the table look like if you add SigLip to Table 3?
* Input-dependence and input-independence in this setting wasn't really defined until much later in the paper than needed (Section 4)
* I don't understand the caption (or value) of Table 1.  What is the average segmentation performance of a DINO or ClIP when trained for that task?  What is state of the art?
* Some examples of the before and after in Table 2 would be nice.  What is state of the art for Table 2?
* In Section 5.5 it's not clear to me whether it's the value tokens or the initial encoder that matters here.  What is the performance of the visual encoder alone?
* Isn't Figure 3 just saying that if you have more parameters, you do better at vision?  What am I missing here?
* Why was the "intervention" done on the particular layers as described?  It seems like there are many ways this could have been done.

**Reasons To Accept:**

Overall, there are potentially nice insights from the paper, though I found it challenging to read at times.  The experiments could be interesting, and understanding the workings of MLMMs is valuable work.

**Reasons To Reject:**

Not discussed in the paper is that MLMs are trained using CLIP and DINO as their visual encoders (via a connector) and that MLMMs have way more parameters, so you’d expect them to be even better at vision tasks, if not just as good.  In some sense, the results aren’t that surprising as a result.  I also didn’t find the PCA outputs (e.g., Figure 8) very readable.

I was particularly surprised that the authors didn't use Siglip as well, as it's what was used in LLaVA Onevision, which they compared against in the paper.

---

> ### Author Response · Authors · 2025-06-03
> **Response to Reviewer #1**
>
> We thank Reviewer #1 for your time and detailed comments. We address them below.
>
> - **Reasons to Reject #1: MLMs are trained using CLIP and DINO as their visual encoders (via a connector) and that MLMMs have way more parameters, so you’d expect them to be even better at vision tasks, if not just as good. In some sense, the results aren’t that surprising as a result.**
>   - We agree that MLMs would be expected to build upon the information provided by their component visual encoder, and should thus perform better than their visual encoder, and MLM intermediate representations (image value tokens). **However, we actually find that the opposite is true:** the visual encoder (SigLIP) outperforms the intermediate values of the MLM on three of four segmentation tasks (details under response to Reasons to Reject #3). Further, the intermediate values actually outperform the overall MLM on the semantic correspondence task (discussed in Section 5.5), showing that the language model does not retain the perceptual information from the visual encoding throughout its layers. We expect that this occurs because the MLM is primarily trained on vision-language tasks such as captioning and VQA, rather than perception-heavy tasks such as those requiring segmentation. These results underscore the need to study visual encodings at different parts of the model, to redirect research towards improving them where needed.
> - **Reasons to Reject #2: I also didn’t find the PCA outputs (e.g., Figure 8) very readable.**
>   - We are happy to modify our figures to be more readable, especially Figure 8, which has double the number of heads compared to the other models (and was hence placed in the Appendix). The main takeaway is that some keys are input-independent, whereas some keys are input-dependent, which is leveraged in our subsequent intervention experiment to mitigate model hallucination.
> - **Reasons to Reject #3: I was particularly surprised that the authors didn't use Siglip as well, as it's what was used in LLaVA Onevision, which they compared against in the paper.**
>   - Thank you for your suggestion. We have obtained results with SigLIP.
>   - On **segmentation**, we find that SigLIP actually beats intermediate representations of LLaVA-OV 7B on three of four segmentation tasks, as shown below:
>    | | Ref. Exp. Seg. | Semantic Seg. | Co-Seg. | Foreground Seg. |
> |---|:---:|:---:|:---:|:---:|
> | SigLIP | 61.7 | 59.1 | 62.0 | 27.1 |
> | LLaVA-OV Intermediate Values | 64.8 | 54.6 | 61.6 | 26.3 |
>   - As discussed under our response to point #1, we believe that this occurs due to the vision-language training of the MLM causing forgetting of perception capabilities through the layers. The only segmentation task in which LLaVA intermediate representations beat SigLIP is Referring Expression Segmentation, which requires more vision-language capabilities.
>   - On **semantic correspondence** (SPair71K), we find that SigLIP performs worse than intermediate representations of LLaVA-OV 7B, as shown below:
> | | Semantic Correspondence |
> |---|:---:|
> | SigLIP | 41.6 |
> | LLaVA-OV Intermediate Values | 46.1 |
>   - This could in part be due to the more semantic nature of the task (similar to referring expression segmentation), and in part due to the fact that LLaVA was trained on video data in addition to images, and is more capable of comparing multiple images.
>   - We will certainly add these numbers to the paper, alongside a discussion on how further research is required to ensure the language model enhances vision encodings rather than degrades them, across various tasks.

---

> ### Author Response · Authors · 2025-06-03
> **Response to Reviewer #1 (continued)**
>
> We now address the Questions to Authors:
>
> - **Questions to Authors #1: Can you please clarify the way that you think about the interaction between visual encoders (SigLip, CLIP, DINO) and MLMMs? And how that factors into the results? I would expect at least a discussion in the paper on this. MLMMs should be strictly better.**
>   - Our work shows that although visual encoders capture perception-related information, *that information is not necessarily retained by the language model layers*, resulting in the MLM outputting incorrect answers on perception-related tasks.
>   - In Section 5.5, we show that in **7.5%** of semantic correspondence inputs, the MLM output was incorrect despite the intermediate MLM value representation being correct. We ran further experiments and find that in **6%** of semantic correspondence inputs, the MLM output was incorrect despite the SigLIP representation being correct.
>   - This makes a strong point calling for further research to ensure the layers of the language model retain perception-related information from the visual encoder. We will add these numbers to the paper, along with a detailed discussion. Thank you for this suggestion.
>
> - **Questions to Authors #2: What does the table look like if you add SigLip to Table 3?**
>   - Thank you for your suggestion. SigLIP obtains **41.6 PCK** on SPair71K, compared to the intermediate representations of LLaVA-OV 7B, which obtain **46.1 PCK** (reported and discussed under Reasons to Reject #1).
>
> - **Questions to Authors #3: Input-dependence and input-independence in this setting wasn't really defined until much later in the paper than needed (Section 4)**
>   - Thank you, we will update the writing to clarify.
>
> - **Questions to Authors #4: I don't understand the caption (or value) of Table 1. What is the average segmentation performance of a DINO or CLIP when trained for that task? What is state of the art?**
>   - Table 1 shows the performance of the image value tokens (i.e., intermediate representations within the language model from the key-value cache) averaged across 4 segmentation tasks. Their performance on each task is reported in Appendix Table 4. They may not reach state-of-the-art, but these results suggest that MLMs should have good perception capabilities — however, we show in Section 5.5 that MLMs do not retain the perception capabilities of their intermediate representations. Our SigLIP results, as discussed above, further show that the language model layers in fact *degrade* perceptual capabilities of the input visual encoding. We will add this discussion to the paper. Thank you for your suggestion.
>
> - **Questions to Authors #5: Some examples of the before and after in Table 2 would be nice. What is state of the art for Table 2?**
>   - Thank you, we will add these examples and results to the paper, and respond further during the discussion period.
>
> - **Questions to Authors #6: In Section 5.5 it's not clear to me whether it's the value tokens or the initial encoder that matters here. What is the performance of the visual encoder alone?**
>   - Thank you for your suggestion. We evaluate the visual encoder SigLIP on the experiment in Section 5.5 and find:
> | | Semantic Correspondence|
> |---|:---:|
> |SigLIP| 37.5% |
> | Intermediate Values | 41% |
> | Overall MLM | 36% |
>
>   - In this task, it is clear that language model layers can initially build upon the perceptual capabilities of the input visual encoding, but then lose it by the time of the MLM output. (in contrast to segmentation, where the language model just degrades the perception). This finding shows the need to focus on researching retention of perception by language model layers. We will add these results to our paper.
>
>  - **Questions to Authors #7: Isn't Figure 3 just saying that if you have more parameters, you do better at vision? What am I missing here?**
>    - Figure 3 shows the correlation between performance of the MLM’s intermediate representations on perception tasks (X-axis) and the performance of the MLM itself on perception tasks (Y-axis). We see a strong correlation between the two, highlighting the importance of studying intermediate representations of MLMs.
>    - To isolate the number of parameters, we also evaluate two models with the same number of parameters (Qwen2-7B and  Qwen2.5-7B), one of which performs better than the other.
>
>
> - **Questions to Authors #8: Why was the "intervention" done on the particular layers as described? It seems like there are many ways this could have been done.**
>   - Thank you for your suggestion. We have conducted further experiments with hallucination intervention, which are presented and discussed in detail in the General Response to Reviewers.
>
>
> Thank you again, and please let us know if you have any questions, or suggestions for further experiments. We will be happy to answer and provide the same.

---

> ### Comment · Area_Chair_kWGv · 2025-06-07
> **Discussion**
>
> Dear Reviewer RSnv,
>
> The authors have responded to your questions. Does their answer address your questions and reasons to reject?

---

> ### Author Response · Authors · 2025-06-09
> **Are there any further questions/concerns we could address?**
>
> We once again thank all the reviewers for their time. We believe we have addressed all questions and concerns in our response, and that our paper will certainly be stronger upon incorporating these new results.
>
> We wanted to follow up in advance of the upcoming deadline — we hope our response may lead you to consider updating your rating of our work.
>
> We have summarized our response and our contributions in "General Response #2" above.
>
> Please let us know if you have any remaining questions or concerns that we could clarify!

---

> ### Comment · Reviewer_RSnv · 2025-06-10
> **Authors' response**
>
> After reading the authors response, I believe that there is still substantial work to be done to disentangle the performance of the visual encoder from the larger MLMM.  As a result, I hold my rating and think this paper could benefit from another round of review.

---

### Author Response · Authors · 2025-06-03
**General Response**

We thank the reviewers for their time, and for highlighting our *“innovative research perspective”*, *“interesting experiments and results”*, *“strong empirical evaluation”*, *“valuable work”* and *“useful work to improve understanding and performance of MLMs”*.

In our general response to all reviewers, we discuss further experiments on the hallucination task (from Section 4.2 of our paper).

In our paper, we calculated variance of image key representations across **1000 COCO images**, then set a threshold to determine input-independent keys and input-dependent keys.

Here, we calculate the variance across the validation set of **ADE20K** (which consists of **2000 images**), then set a threshold.

We find that when plotting the variance of each key, a **bimodal distribution** forms:
- First mode centered at variance of **200**.
- Second mode centered at variance of **600**.

We set the threshold at the lowest point between these two modes, at a variance of **450**.
This pattern was consistent between ADE20K and COCO, addressing **R3’s question** about the image distribution.

We find that when setting this threshold:
- **20 keys** are input-independent in the first 8 layers (including all 16 keys in the first four layers),
- **8 keys** are input-independent in the next 10 layers,
- **14 keys** are input-independent in the last 10 layers.

We call these the *early layers*, *middle layers* and *late layers* respectively.

We then run the intervention by blocking:
1. Only the input-independent keys at early layers;
2. Only the input-independent keys at middle layers;
3. Only the input-independent keys at late layers,

in accordance with our hypothesis (from Darcet et al. 2024) that input-independence at middle-to-later layers could encode artifacts.

Base model performance: 78.1 F1 on POPE
Intervening on input-independent keys at early layers: 65.1 F1 on POPE
Intervening on input-independent keys at middle layers: 77.9 F1 on POPE
Intervening on input-independent keys at late layers: 81.2 F1 on POPE

Our results show that input-independent keys do encode artifacts at later layers, and that this finding can be leveraged to mitigate hallucination in MLMs.

Please let us know if you have any questions, or suggestions for further experiments. We will be happy to answer and provide the same.

---

### Author Response · Authors · 2025-06-09
**Following up with Reviewers (General Response #2)**

We once again thank all the reviewers for their time. **We believe we have addressed all questions and concerns in our response, and that our paper will certainly be stronger upon incorporating these new results.**

We wanted to follow up in advance of the upcoming deadline — **we hope our response may lead you to consider updating your rating of our work.** Please let us know if you have any remaining questions or concerns that we could clarify.

-----

- To summarize, the main points of concern about our work were:
  1. **Concern 1 [Reviewers 1, 2 and 3]**: Further numbers needed for the hallucination experiment: why were the specific thresholds and layers chosen?
  2. **Concern 2 [Reviewers 1 and 2]**: Performance of intermediate VLM representations (image value tokens) compared to vision encoders like SigLIP.

- **We addressed Concern 1 with additional experimental results** (as discussed in the General Response to Reviewers), definitively showing that (1) the image key tokens at later layers of VLMs indeed encode artifacts; and (2) these tokens can be leveraged to mitigate hallucination. We discuss details about different thresholds and layers in the response.
- **We addressed Concern 2 by highlighting some results in the paper, as well as additional experimental results** comparing intermediate VLM representations (image value tokens) to SigLIP on a variety of tasks: four different segmentation tasks, as well as the semantic correspondence task. These results address the concern that VLMs “should” outperform SigLIP: *the opposite is indeed true*, where SigLIP tends to outperform intermediate VLM representations (per our new results), which in turn outperform the VLM text output (per Section 5.5). These results thus deeply underscore the need for research efforts to be directed towards how language model layers process visual encodings, which could significantly improve VLM perception capabilities.

-----

- **Overall, our main contributions are:**
  1. Carrying out the first systematic study of key-value representations in VLMs, revealing the KV cache as an important focus of analysis of image processing by language models (as the image KV tokens serve as a proxy for the image through the layers of the language model).
  2. Exposing the tendency of image key tokens in later layers to encode artifacts, a fact that can be leveraged to mitigate hallucination in VLMs.
  3. Highlighting the perception capabilities of intermediate image value tokens in VLMs via their performance on perception-heavy tasks, and further revealing that these capabilities are lost through the language model layers, resulting in the current “poor” perception of VLMs.
- **Our work reveals key areas for future research to focus on, in order to improve VLM performance on a variety of perception-heavy tasks,** including various types of segmentation, referring expression detection, hallucination, semantic correspondence and temporal correspondence.

---

### Author Response · Authors · 2025-06-10
**Results for Section 4.2 (hallucination intervention) on more models and benchmarks (General Response #3)**

We again thank all the reviewers for their time, and for their follow-up questions. **Reviewer #2 (SMbp) and #3 (GABW) have both requested results of our intervention experiment in Section 4.2 on additional benchmarks, and R2 requested results from an additional model. We provide these results below.**

**First, we re-ran the hallucination experiment on POPE with a new model, Qwen2.5-VL**. For this model, we again calculate variance of image key tokens across 1000 COCO images, and set a threshold at 450 (the lowest point between the two modes of the variance distribution, centered at 200 and 625 respectively). When setting this threshold, 20 keys are input-independent in the first 8 layers, 6 keys are input-independent in the next 10 layers, and 14 keys are input-independent in the last 10 layers. We call these the “early layers”, “middle layers” and “late layers” respectively.

As in our previous experiments (in General Response #1), we then run the intervention by blocking (1) only the input-independent keys at early layers; and (2) only the input-independent keys at middle layers; and (3) only the input-independent keys at late layers, in accordance with our hypothesis (from Darcet et al 2024) that input-independence at middle-to-later layers could encode artifacts.

Results of Qwen2.5-VL on POPE are below, along with LLaVA-OV on POPE per General Response #1:
| Model | Which input-independent keys were blocked | F1 on POPE |
|---|---|:---:|
| Qwen2.5-VL | none (base model) | 87.9 |
|  | early layers | 69.1 |
|  | middle layers | 82.4 |
|  | late layers | **89.2** |
| | | |
| LLaVA-OV | none (base model) | 78.1 |
|  | early layers | 65.1 |
|  | middle layers | 77.9 |
|  | late layers | **81.2** |
| | | |

**These results clearly show the applicability of our findings to multiple models**: specifically, the existence of input-independent image key tokens in later layers of the model, which (similar to the finding from Darcet et al 2024) encode artifacts, and intervening on the same to mitigate hallucination on POPE.

-------

**Next, we show results on another popular hallucination benchmark: MME** (Fu et al 2024). Following the setup from "Self-Correcting Decoding with Generative Feedback for Mitigating Hallucinations in Large Vision-Language Models" (Zhang et al, ICLR 2025), we evaluate on these subsets of MME: count, color, position and existence. In all four of these, the model is asked a Yes/No question about the image (similar to POPE), **which address hallucination across various axes and also measure general capabilities (addressing R3's concern).**

We follow the same setup as the POPE experiment, blocking model-specific input-independent image key tokens at "late" layers. The results on Qwen2.5-VL and LLaVA-OV are below (Score measured out of 200 points, per the MME setup):

| Model | MME subset | Score of base model | Score after intervention |
|---|---|:---:|:---:|
| Qwen2.5-VL | count | **182.5** | **182.5** |
| | color | 185.8 | **193.3** |
| | position  | 160.0 | **165.8** |
| | existence | **200.0** | **200.0** |
| | | |
| LLaVA-OV | count | 170.0 | **175.0** |
| | color | 178.3 | **187.5** |
| | position | 145.0 | **153.3** |
| | existence | **200.0** | **200.0** |
| | | |

Note: the "existence" subset, which is the closest equivalent to POPE, is much smaller than POPE (60 data points compared to 9000 in POPE). The models seem to achieve perfect existence scores both before and after intervening, hinting that this existence subset is both smaller and easier than POPE.

**These results show that our findings are also applicable across multiple benchmarks**: specifically, that intervening on input-independent image key tokens can improve (or at least preserve) performance of the model across various benchmarks.

----

**We thank Reviewers 2 and 3 for their suggestions. These results strengthen the first two contributions of our paper as listed in General Response #2, showing them to be applicable to multiple models and benchmarks**.

**Further, our third contribution has been demonstrated across 6 separate benchmarks/tasks** (few-shot foreground segmentation, co-segmentation, semantic segmentation, referring expression segmentation, semantic correspondence and temporal correspondence): highlighting the perception capabilities of intermediate image value tokens in VLMs via their performance on perception-heavy tasks, and further revealing that these capabilities are lost through the language model layers, resulting in the current “poor” perception of VLMs.

**We believe this addresses the remaining concerns of Reviewers 2 and 3, and that our contributions together serve as an important message to the research community on the need to improve language model processing of visual tokens.** Please let us know if there are any further questions, we would be happy to clarify. We look forward to hearing your response!

---

### Decision · Program_Chairs · 2025-07-07

**Decision:**

Accept

**Comment:**

This paper focuses on the key-value representations corresponding to image tokens in 3 multimodal language models (MLMs). The authors locate keys that are deemed to be independent of the input image (based on variance) and find that blocking the attention going to these keys (particularly in middle-to-later layers) leads to a decrease in hallucinations in object identification. The study then reveals that the MLMs are not able to utilize the information encoded in the value tokens for perception-heavy tasks such as segmentation, semantic correspondence and referring expression detection. The outcomes indicate the importance of key-value tokens for understanding the information flow in MLMs as they process visual inputs.

**Pros**

Quality: The reviewers appreciated the extensive experiments and appropriate analyses.

Originality: The paper offers unique insights on visual key-value tokens in MLMs in contrast to work exploring vision transformer-based encoders and the outputs of transformer blocks.

Significance:  This is an interesting paper that uncovers significant features of visual key-value tokens, and the consequences of altering them, with implications for improving perception-centric performance, interpretability and hallucination reduction.

**Cons**

The paper could benefit from clarifying the claims and improving the presentation of tables, figures, symbols and terms.

One main question was whether blocking the attention to keys that are deemed to be input independent would have similar outcomes in general-purpose benchmarks. The authors claim that their results reflect general capabilities, listing tasks such as semantic segmentation and referring expression segmentation as well as including results from an extra hallucination benchmark that covers multiple axes such as color, position and existence. Reviewers were not very convinced of the claims regarding the implications for general-purpose tasks, indicating that some claims lack experimental support.

The authors ran further intervention tests on a larger subset from another dataset, evaluated another vision encoder, and provided results on an additional hallucination benchmark. I suggest incorporating these in the paper.

I think the responses were reasonable and adequately addressed the majority of the concerns. The findings would be valuable as a focused contribution on vision-centric tasks, hinting at ways to mitigate hallucination as well as improve performance in MLMs.